# ATP binding facilitates target search of SWR1 chromatin remodeler by promoting one-dimensional diffusion on DNA

Claudia C Carcamo[1], Matthew F Poyton[1], Anand Ranjan[2], Giho Park[2], Robert K Louder[2], Xinyu A Feng[1], Jee Min Kim[2], Thuc Dzu[2], Carl Wu[2]*, Taekjip Ha[1,3,4,5]*

[1]Department of Biophysics and Biophysical Chemistry, Johns Hopkins University, Baltimore, United States; [2]Department of Biology, Johns Hopkins University, Baltimore, United States; [3]Howard Hughes Medical Institute, Baltimore, United States; [4]Johns Hopkins University, Department of Biomedical Engineering, Baltimore, United States; [5]Johns Hopkins University, Department of Biophysics, Baltimore, United States

*For correspondence:
wuc@jhu.edu (CW);
tjha@jhu.edu (TH)

**Abstract** One-dimensional (1D) target search is a well-characterized phenomenon for many DNA-binding proteins but is poorly understood for chromatin remodelers. Herein, we characterize the 1D scanning properties of SWR1, a conserved yeast chromatin remodeler that performs histone exchange on +1 nucleosomes adjacent to a nucleosome-depleted region (NDR) at gene promoters. We demonstrate that SWR1 has a kinetic binding preference for DNA of NDR length as opposed to gene-body linker length DNA. Using single and dual color single-particle tracking on DNA stretched with optical tweezers, we directly observe SWR1 diffusion on DNA. We found that various factors impact SWR1 scanning, including ATP which promotes diffusion through nucleotide binding rather than ATP hydrolysis. A DNA-binding subunit, Swc2, plays an important role in the overall diffusive behavior of the complex, as the subunit in isolation retains similar, although faster, scanning properties as the whole remodeler. ATP-bound SWR1 slides until it encounters a protein roadblock, of which we tested dCas9 and nucleosomes. The median diffusion coefficient, 0.024 μm$^2$/s, in the regime of helical sliding, would mediate rapid encounter of NDR-flanking nucleosomes at length scales found in cellular chromatin.

## Editor's evaluation

This manuscript provides exciting and timely new insight into the target search mechanism of SWR1, with fundamentally important implications for other nucleosome remodelers. The authors use extremely elegant single-molecule approaches to study the interaction of SWR1 with DNA. This study provides the first conclusive demonstration that SWR1 undergoes 1D diffusion along DNA, which plays an important role in finding the correct nucleosomal substrate in vivo by guiding SWR1 molecules that bind to nucleosome-depleted regions towards flanking nucleosomes.

## Introduction

Eukaryotic genomes are packaged into chromatin, the base unit of which is the nucleosome. Both the position of nucleosomes on the genome and their histone composition are actively regulated by chromatin remodeling enzymes (*Yen et al., 2012*). These chromatin remodelers maintain and modify chromatin architecture which regulates transcription, replication, and DNA repair (*Tessarz*

*and Kouzarides, 2014*). A particularly well-defined area of chromatin architecture is found at gene promoters in eukaryotes: a nucleosome-depleted region (NDR) of about 140 bp in length is flanked by two well-positioned nucleosomes, one of which, the +1 nucleosome, sits on the transcription start site (TSS) (*Bernstein et al., 2004*; *Lee et al., 2007*; *Xu et al., 2009*; *Yuan et al., 2005*) and the nucleosome on the opposite side of the NDR, upstream of the TSS, is known as the −1 nucleosome. The +1 nucleosome is enriched for the noncanonical histone variant H2A.Z (*Albert et al., 2007*; *Raisner et al., 2005*) and, in yeast, H2A.Z is deposited into the +1 nucleosome by SWR1 (Swi2/Snf2-related ATPase complex), a chromatin remodeler in the INO80 family of remodelers (*Korber and Hörz, 2004*; *Krogan et al., 2004*; *Mizuguchi et al., 2004*; *Ranjan et al., 2013*). The insertion of H2A.Z into the +1 nucleosome is highly conserved and plays an important role in regulating transcription (*Giaimo et al., 2019*; *Rudnizky et al., 2016*).

While the biochemistry of histone exchange has been characterized, the target search mechanism SWR1 uses to preferentially exchange H2A.Z into the +1 nucleosome is not yet understood. The affinity of SWR1 for nucleosomes is enhanced by both long-linker DNA (*Ranjan et al., 2013*; *Yen et al., 2013*) and histone acetylation (*Zhang et al., 2005*), and both factors play a role in the recruitment of SWR1 to promoters. A recent single-molecule study further showed that SWR1 likely exploits preferential interactions with long-linker length DNA by demonstrating that H2A.Z is predominantly deposited on the long-linker distal face of the nucleosome (*Poyton et al., 2022*), similar to what is observed in vivo (*Rhee et al., 2014*). It is possible that SWR1 first binds long-linker DNA and then finds its target, the +1 nucleosome, using facilitated diffusion (*Figure 1A*), as was previously suggested (*Ranjan et al., 2013*). In a hypothetical facilitated search process SWR1 would first find the NDR through a three-dimensional target search. Once bound, it is possible the entire SWR1 complex diffuses one dimensionally on the NDR, where it can encounter both the −1 and +1 nucleosomes.

Facilitated diffusion has been shown to be essential for expediting the rate at which transcription factors and other DNA-binding proteins can bind their target compared to a 3D search alone (*Berg et al., 1981*; *Elf et al., 2007*; *Hannon et al., 1986*; *Ricchetti et al., 1988*; *von Hippel and Berg, 1989*). Recently published in vivo single-particle tracking found that chromatin remodelers have bound-state diffusion coefficients that are larger than that of bound H2B, hinting at the possibility that they may scan chromatin, but those studies could not distinguish between remodeler scanning and locally enhanced chromatin mobility (*Kim et al., 2021*; *Ranjan et al., 2020*). It is not known if SWR1 or any other chromatin remodeler can linearly diffuse on DNA, and therefore make use of facilitated diffusion to expedite its target search process. Additionally, SWR1's core ATPase, like other chromatin remodelers, is a superfamily 2 (SF2) double-stranded DNA translocase (*Nodelman and Bowman, 2021*; *Yan and Chen, 2020*). While there is no evidence for SWR1 translocation on nucleosomal DNA, it remains possible that SWR1 may undergo directed, instead of diffusional, movements on a DNA duplex in the absence of a nucleosome substrate.

In this study, we used a site-specifically labeled SWR1 complex to demonstrate that in vitro SWR1 can scan long stretches of DNA via 1D diffusion, which may facilitate its search for target nucleosomes in vivo. First, we characterized the kinetics of SWR1 binding to DNA and found that the on-rate increases linearly with DNA length while the off-rate is independent of length for DNA longer than 60 bp. Next, we used an optical trap equipped with a scanning confocal microscope to show that SWR1 can diffuse one dimensionally along stretched DNA, with a diffusion coefficient that permits scanning of a typical NDR in 93 ms. Interestingly, we see that ATP binding alone increases the one-dimensional diffusion coefficient of SWR1 along DNA. We found that a major DNA-binding subunit of the SWR1 complex, Swc2, also diffuses on DNA suggesting that it contributes to SWR1's diffusivity. The diffusion coefficient for both SWR1 and Swc2 increases with ionic strength suggesting that SWR1 utilizes some microscopic dissociation and reassociation events, known as hopping, to diffuse on DNA. However, it is likely that SWR1 only makes infrequent hops, with most of the diffusion on DNA being mediated by helically coupled diffusion, known as sliding, since SWR1 diffusion is blocked by proteins that are bound to DNA, such as dCas9, and the diffusion of the complex is slower than would be expected for majority hopping diffusion. Finally, we observed SWR1 diffusion on DNA containing sparsely deposited nucleosomes and provide evidence that SWR1 diffusion is largely confined between nucleosomes. Our data indicate that a multisubunit chromatin remodeler can diffuse along DNA and suggests that in vivo SWR1 may find NDR-flanking nucleosomes through facilitated diffusion. 1D diffusion at NDRs may be an important aspect of SWR1 target search and

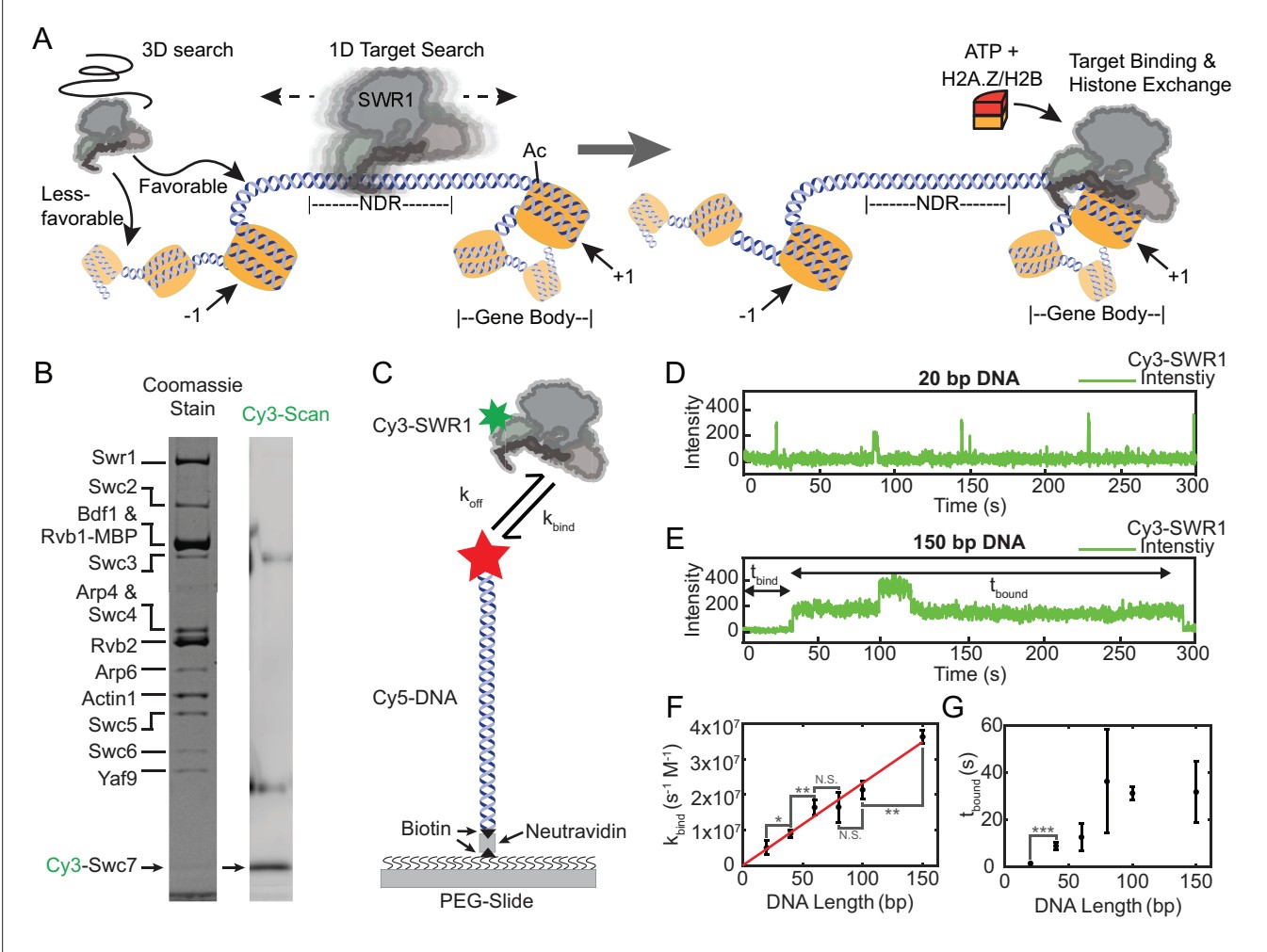

**Figure 1.** SWR1 binds DNA in short- and long-lived states and prefers longer DNAs. (**A**) Proposed facilitated search mechanism for how SWR1 locates the +1 nucleosome. (**B**) A denaturing sodium dodecyl sulfate–polyacrylamide gel electrophoresis (SDS–PAGE) of reconstituted Cy3-SWR1 imaged for Coomassie (left) and Cy3 fluorescence (right). Cy3-Swc7 is faint when stained with Coomassie but is a prominent band in the Cy3 scan. The two diffuse bands that run at higher molecular weight and appear in the Cy3 scan are carry over from the ladder loaded in the adjacent lane. (**C**) A schematic of the single-molecule colocalization experiment used for kinetic measurements of Cy3-SWR1 binding to Cy5-labeled DNA of different lengths. Representative traces of Cy3-SWR1 binding to (**D**) 20 bp Cy5-DNA and to (**E**) 150 bp Cy5-DNA. A second Cy3-SWR1 can be seen binding at approximately 100 s. (**F**) The on-rate binding constant for the initial binding event, ($k_{bind}$) for SWR1 to DNA of different lengths, error bars are standard deviation. *N* values: 20 bp (40), 40 bp (67), 60 bp (267), 80 bp (129), 100 bp (221), 150 bp (409). The red line is a linear fit to the data, where $R^2 = 0.99$ [two technical replicates represented, statistical differences determine via Student's *t*-test, where asterisks indicate the level of significance as conventionally defined (* = $p < 0.05$; ** = $p < 0.01$; *** = $p < 0.001$, n.s. = not significant)]. (**G**) The lifetime ($t_{bound}$) of Cy3-SWR1 bound to DNAs of different lengths, error bars are standard deviation. *N* values: 20 bp (48), 40 bp (118), 60 bp (291), 80 bp (382), 100 bp (339), 150 bp (363) [two technical replicates represented, statistical differences determine via Student's *t*-test, where asterisks indicate the level of significance as conventionally defined].

The online version of this article includes the following source data and figure supplement(s) for figure 1:

**Source data 1.** Numerical data and statistics underlying panels F and G.

**Source data 2.** Gel images (Coomassie and Cy3 scans) shown in panel B.

**Figure supplement 1.** Cy3-SWR1 purification and DNA-binding kinetics.

**Figure supplement 1—source data 1.** Gel images (Coomassie and Cy3 scans) shown in panels A and B.

**Figure supplement 1—source data 2.** Excel file corresponding to panels C, F, and H.

potentially a common feature for chromatin remodelers that acts upon nucleosomes adjacent to the NDR.

## Results

### SWR1-binding kinetics depend on DNA length

To study both the DNA-binding kinetics and diffusive behavior of SWR1, we generated a site-specifically labeled complex referred to as Cy3-SWR1 (*Figure 1B*). We purified SWR1 from *Saccharomyces cerevisiae* in the absence of the Swc7 subunit (SWR1ΔSwc7). Recombinant Swc7 was expressed and purified from *Escherichia coli*, a single cysteine in Swc7 was labeled with Cy3, and the labeled Swc7 was then added to the SWR1ΔSwc7 preparation between two steps of a specialized tandem affinity purification protocol (*Sun et al., 2020*). Subsequent purification on a glycerol gradient revealed that the Cy3-labeled Swc7 comigrated with the rest of the SWR1 subunits, demonstrating incorporation of Swc7 back into the SWR1 complex (*Figure 1—figure supplement 1A*). The histone exchange activity of the labeled Cy3-SWR1 was identical to that of wild-type SWR1 as revealed by an electrophoretic mobility shift assay (EMSA) which shows the histone exchange reaction progress as triple FLAG-tagged H2A.Z is incorporated into a mononucleosome substrate (*Figure 1—figure supplement 1B*).

While it is well established that the affinity of SWR1 for DNA is dependent on DNA length (*Ranjan et al., 2013*), the kinetics of binding are unknown. We used TIRF (Total Internal Reflection Fluorescence) microscopy to perform single-molecule colocalization measurements to observe Cy3-SWR1 binding and unbinding on Cy5-labeled DNA of different lengths in real time (*Figure 1C–G*). These measurements showed that both the on-rate ($k_{bind}$) and the lifetime of the SWR1–DNA complex ($t_{bound}$) are dependent on DNA length. The on-rate for SWR1 binding to 20 bp DNA, the approximate size of linker DNA between intragenic nucleosomes in yeast, was $1 \times 10^6$ M$^{-1}$ s$^{-1}$. Increasing the DNA length to 150 bp, the approximate size of the NDR in yeast, significantly increased the binding rate 36-fold to $3.6 \times 10^7$ M$^{-1}$ s$^{-1}$. $k_{bind}$ increased linearly with DNA length between these two values (*Figure 1F*), indicating that as the effective concentration of DNA bp is increased, binding is also increased (*Figure 1—figure supplement 1C*; $k_{bind}$ normalized by DNA length). This result suggests that longer DNA substrates may harbor more potential SWR1-binding sites. In support of this, we found that longer DNA molecules could accommodate multiple bound SWR1 molecules, with the likelihood of multiple binding events increasing with DNA length (see *Figure 1E* for an example trace). Cy3-Swc7 alone exhibited no affinity for 150 bp DNA (data not shown), suggesting that the observed Cy3-signal increase is caused by the full Cy3–SWR1 complex binding to DNA.

The lifetime of SWR1 bound to DNA ($t_{bound}$) was also sensitive to DNA length, exhibiting two sharp increases as DNA size increased from 20 to 40 and 60 to 100 bp. Whereas $t_{bound}$ for 20 bp DNA was 1.5 ± 0.3 s, $t_{bound}$ for SWR1 binding to 40 and 60 bp DNA increased to 9 ± 1.4 and 12 ± 5.8 s, respectively, which is the same within error (*Figure 1G*). Once the DNA was 80 bp or longer, however, the lifetime increased dramatically to at least 30 s, the photobleaching limit of our measurement (*Figure 1—figure supplement 1D*). Measurements at lower laser power showed that SWR1 remained bound to 150 bp DNA with a life time of approximately 3 min (*Figure 1—figure supplement 1D*). $t_{bound}$ was unchanged in the presence of ATP but was sensitive to ionic strength, decreasing with added salt (*Figure 1—figure supplement 1E, F*). Curiously, $t_{bound}$ also decreased in the presence of competitor DNA (*Figure 1—figure supplement 1E, F*). In our in vitro experimental setup, as DNA length is increased, SWR1 $t_{bound}$ showed an approximately 120-fold increase. Compared to the 36-fold increase in $k_{bind}$ as DNA length increased, the much higher fold increase in binding lifetime might suggest that once SWR1 binds NDR DNA in vivo that it will remain bound for several minutes, potentially sequestering the remodeler from performing histone exchange at other +1 nucleosome targets. The in vivo $t_{bound}$, however, is likely much shorter due to the higher ionic strength in cells as well as due to effects of molecular crowding, competitor DNA binding, and the activity of endogenous helicases which may oust DNA-bound factors such as SWR1. Indeed, a study utilizing single-particle tracking in vivo showed that the stable chromatin-bound dwell-time for a number of ATP-dependent remodelers is on the order of several seconds (*Kim et al., 2021*). The kinetic measurements presented here show that the affinity of SWR1 for DNA greater than 60 bp is primarily limited by the on-rate, suggesting the increased occupancy of SWR1 at longer NDRs observed in yeast (*Ranjan et al., 2013*) is a result of

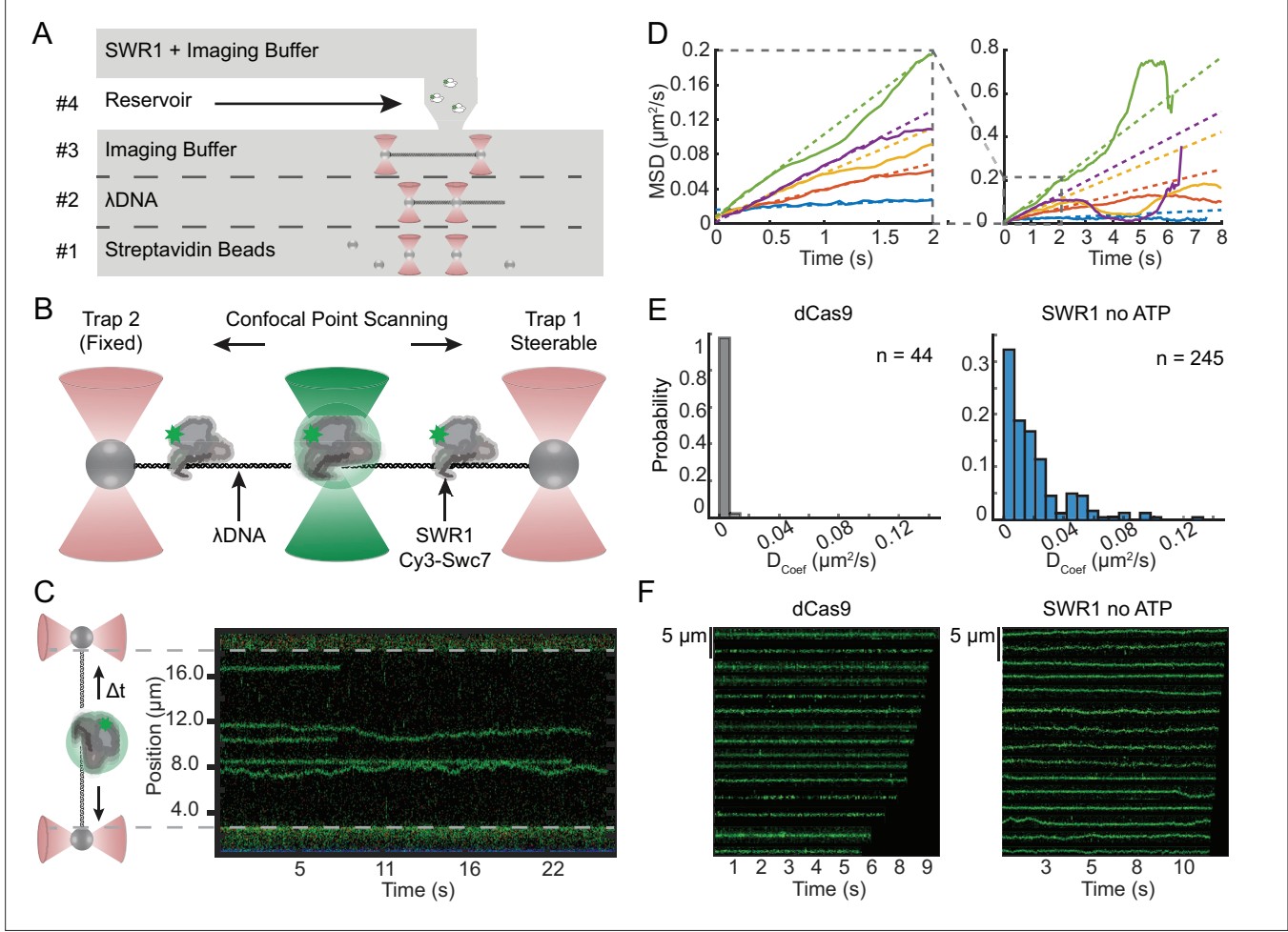

**Figure 2.** SWR1 diffuses on extended dsDNA. (**A**) Schematic representation of a C-Trap microfluidics imaging chamber with experimental workflow depicted therein: #1 catch beads, #2 catch DNA, #3 verify single tether, and #4 image SWR1 bound to DNA. (**B**) Schematic representation of confocal point scanning across the length of lambda DNA tethered between two optically trapped beads. This method is used to monitor the position of fluorescently labeled SWR1 bound to DNA. (**C**) Example kymograph with a side-by-side schematic aiding in the interpretation of the kymograph orientation. (**D**) Mean squared displacement (MSD) versus time for a random subset of SWR1 traces in which no ATP is added. An enlargement of the initial linear portion is shown to the left where colored dashed lines are linear fits to this portion. (**E**) Histogram of diffusion coefficients for dCas9 (left) and SWR1 in which no ATP is added (right). (**F**) Segmented traces of dCas9 (left) and SWR1 in which no ATP is added (right).

The online version of this article includes the following source data and figure supplement(s) for figure 2:

**Source data 1.** Data underlying panels D and E.

**Source data 2.** Uncropped kymograph Tiff image from panel C.

**Figure supplement 1.** Cy3-Swc7 does not bind DNA without the SWR1 complex.

**Figure supplement 1—source data 1.** Raw scans and kymograph tiff files.

---

the increased probability of SWR1 finding the NDR, as opposed to an increase in the residence time of SWR1.

## SWR1 scans DNA

To determine if SWR1 can move along DNA, we tracked single Cy3–SWR1 complexes bound to stretched lambda DNA using an optical trap equipped with a confocal scanning microscope (LUMICKS, C-Trap) (*Heller et al., 2014a*; *Heller et al., 2014b*). The experiment was carried out using a commercial flow cell in order to efficiently catch beads, trap DNA, and image-bound proteins over time (*Figure 2A*) as has been performed previously (*Balaguer et al., 2021*; *Brouwer et al., 2016*; *Gutierrez-Escribano et al., 2019*; *Newton et al., 2019*; *Rill et al., 2020*; *Wasserman et al., 2019*).

Briefly, lambda DNA end-labeled with biotin is tethered between two optically trapped streptavidin-coated polystyrene beads, pulled to 5 pN tension to straighten the DNA (*Baumann et al., 2000*) and the distance between the two optical traps is clamped (*Figure 2A, B*). After confirming the presence of a single DNA tether, the DNA is brought into an adjacent channel of the flow cell containing 250 pM Cy3-SWR1. Confocal point scanning across the length of the DNA was used to image single Cy3-SWR1 bound to lambda DNA over time to generate kymographs (*Figure 2B, C*). The observed fluorescent spots represent the Cy3–SWR1 complex as Cy3-Swc7 alone was unable to bind DNA (*Figure 2—figure supplement 1*).

Cy3-SWR1 bound to lambda DNA is mobile, demonstrating that Cy3-SWR1 can move on DNA once bound and the movement did not appear to be unidirectional. Therefore, we plotted mean square displacement (MSD) versus time and found that the initial portion of the curve is linear, suggesting that diffusion is Brownian (*Figure 2D*). The diffusion coefficient observed ($D_{1,obs}$) for Cy3-SWR1 was 0.013 ± 0.002 µm²/s in buffer alone (*Figure 2E, F*). Since diffusion coefficient distributions are nonnormal, $D_{1,obs}$ is defined as the median diffusion coefficient of all molecules in a condition; individual diffusion coefficients were determined from the slope of the initially linear portion of their respective MSD plot (see Materials and methods for more details). This diffusion coefficient is comparable to other proteins with characterized 1D diffusion (*Gorman et al., 2007*; *Park et al., 2021*). In contrast $D_{1,obs}$ for specifically bound Cy5-dCas9, an immobile reference with $D_{1,obs}$ of 0.0003 ± 0.0004 µm²/s, is 40-fold smaller than Cy3-SWR1. These measurements clearly show that SWR1 undergoes Brownian diffusion on nucleosome-free DNA.

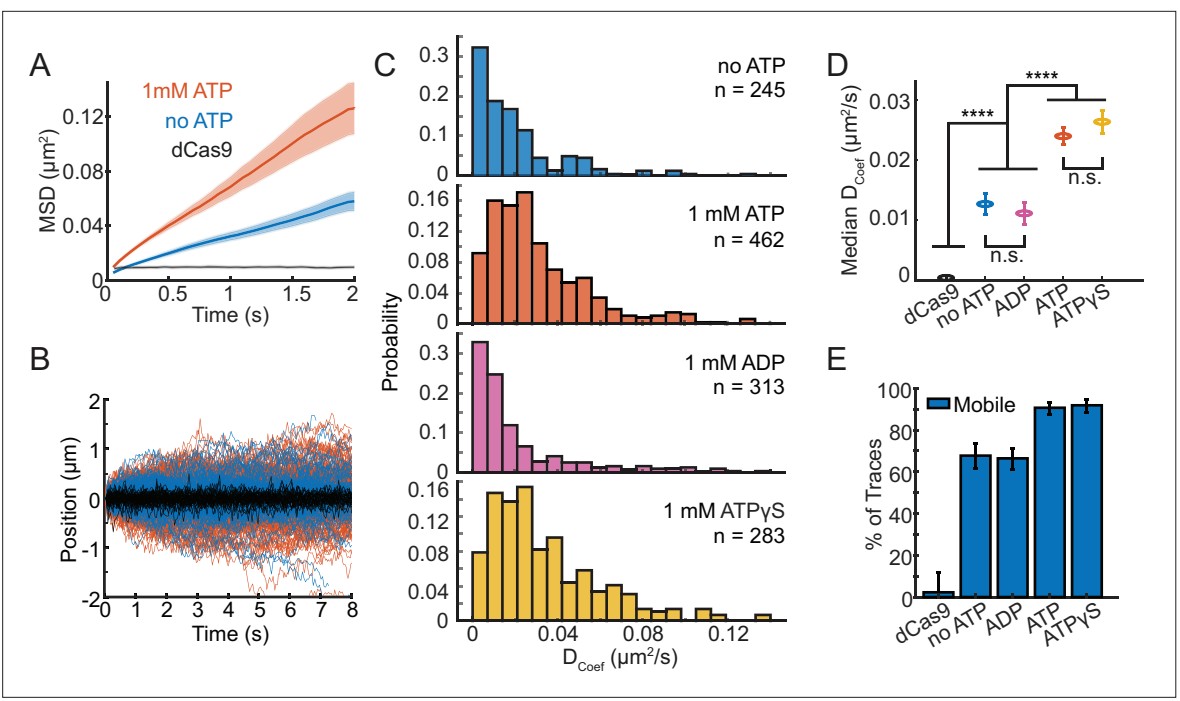

**Figure 3.** ATP binding modulates SWR1 diffusion. (**A**) Mean squared displacement (MSD) versus time plotted for 1 mM ATP (orange, *n* = 124), no ATP (blue, *n* = 134), and dCas9 (black, *n* = 25) with shaded error bars standard error of the mean (SEM). (**B**) SWR1 trajectories aligned at their starts for 1 mM ATP (orange lines), no ATP (blue lines), and dCas9 as reference for immobility (black lines). All trajectories represented. (**C**) Histograms of diffusion coefficients extracted from individual trajectories for SWR1 diffusion in the presence of no ATP, 1 mM ATP, 1 mM ADP, and 1 mM ATPγS (from top to bottom). The number of molecules measured (*n*) for each condition is printed in each panel. (**D**) Median diffusion coefficients for SWR1 in varying nucleotide conditions. dCas9 is shown as a reference. Error bars are the uncertainty of the median. Statistical differences were determined via Student's *t*-test, and asterisks indicate the level of significance (**** = p < 0.0001, n.s. = not significant). (**E**) Percentage of mobile traces in each condition, where immobility is defined as traces with similar diffusion coefficients to dCas9 (defined as diffusion coefficients smaller than 0.007 µm²/s). Error bars represent confidence intervals estimated using Matlab 'bootci' run with default settings and with a number of bootstrap samples = 5000, conditions with nonoverlapping error can be considered significantly different.

The online version of this article includes the following source data for figure 3:

**Source data 1.** Numerical data underlying panels A and C–E.

## ATP-bound SWR1 is more diffusive than the unbound complex

To determine if SWR1 can actively translocate on DNA, we observed the motion of Cy3-SWR1 in the presence of 1 mM ATP (*Figure 3*). The MSDs of Cy3-SWR1 in the presence of ATP remained linear, showing that SWR1 does not translocate directionally on DNA (*Figure 3A*). The increased slope of the MSDs in the ATP condition, however, does indicate that ATP increases the diffusion. An overlay of all trajectories from both conditions further demonstrates that SWR1 diffuses a greater distance from the starting position in the presence of ATP and that its motion is not directional (*Figure 3B*). To address whether this increased diffusion was due to ATP hydrolysis, we also measured SWR1 diffusion in the presence of 1 mM ATPγS, a nonhydrolyzable analog of ATP, as well as with ADP. The distribution in diffusion coefficients in the presence of ATP and ATPγS are both shifted to higher values compared to in the absence of ATP or in the presence of ADP (*Figure 3C*). This shift was shown to be statistically significant using the nonparametric Mann–Whitney *U*-test (*Figure 3D*). SWR1 diffusion in the presence of 1 mM ATP ($D_{1,obs} = 0.024$ μm²/s ± 0.001) was not significantly different than diffusion in the presence of 1 mM ATPγS ($D_{1,obs} = 0.026$ μm²/s ± 0.002). Similarly, SWR1 diffusion in the absence of ATP ($D_{1,obs} = 0.013$ μm²/s ± 0.002) was not different than SWR1 diffusion in the presence of 1 mM ADP ($D_{1,obs} = 0.011$ μm²/s ± 0.002). Additionally, we found that ATP decreased the fraction of slow or immobile Cy3-SWR1 molecules, defined as those molecules that show $D_1$ values that are indistinguishable from dCas9

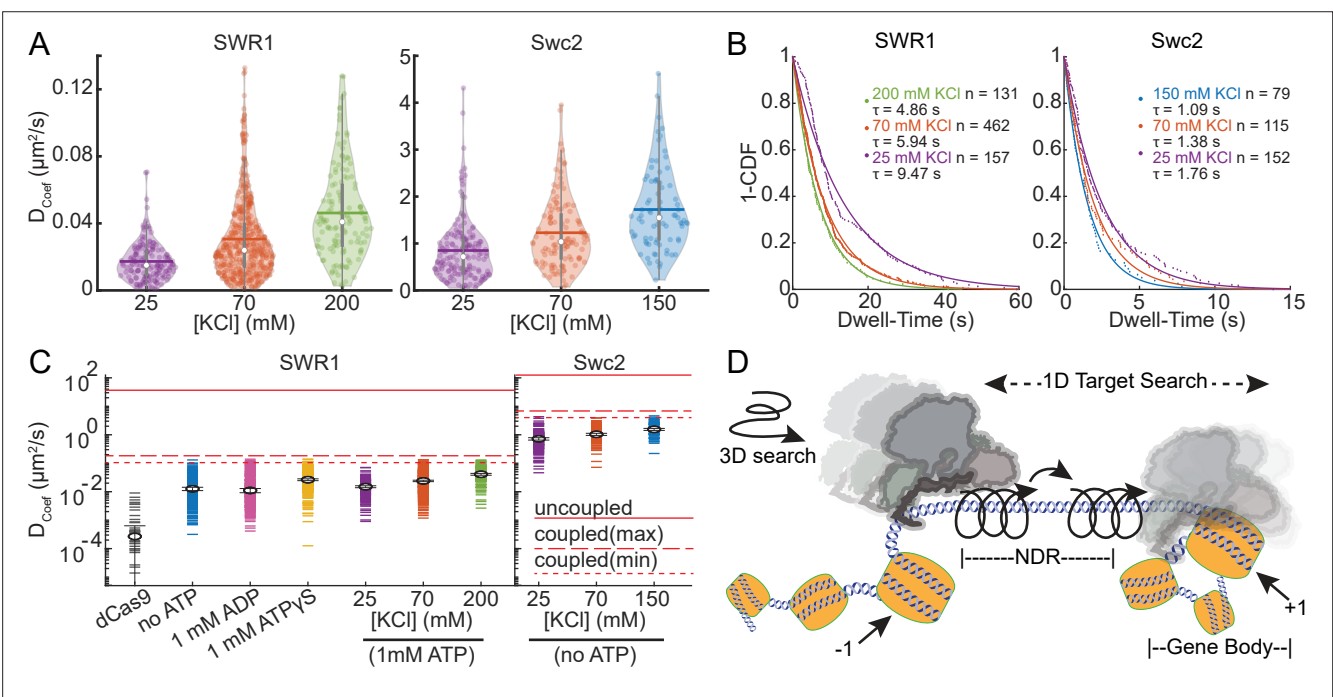

**Figure 4.** SWR1 and Swc2 DNA-binding domain (DBD) utilize a combination of sliding and hopping to scan DNA. (**A**) Violin plots of diffusion coefficients for SWR1 and Swc2 DBD in increasing potassium chloride concentrations. Medians are shown as white circles and the mean is indicated with a thick horizontal line. 70 mM KCl represents the standard salt condition. (**B**) 1-CDF plots of SWR1 and Swc2 were fit to exponential decay functions to determine half-lives of binding in varying concentrations of potassium chloride. The number of molecules as well as half-lives determined is printed therein. Dots represent data points, while solid lines represent fits. Half-lives are calculated using the length of all the trajectories in each condition. (**C**) Upper limits for diffusion of SWR1 and Swc2 predicted using either a helically uncoupled model for hopping diffusion (uppermost solid red line) or a helically coupled model for sliding diffusion (lower dashed red lines). Two dashed lines are shown for helically coupled upper limits because the distance between the helical axis of DNA and the center of mass of either SWR1 or Swc2 is unknown. Markers represent median values. $D_{coef}$ values for each condition are shown as horizontal dashes, the number of molecules represented in each condition is as aforementioned. (**D**) A schematic representation of a model for how SWR1 likely performs 1D diffusion on DNA.

The online version of this article includes the following source data and figure supplement(s) for figure 4:

**Source data 1.** Data underlying panels A–C.

**Figure supplement 1.** Purification and fluorophore labeling of the Swc2 DNA-binding domain (DBD).

**Figure supplement 1—source data 1.** Gel images (Coomassie and Cy3 scans).

**Figure supplement 2.** Rotation-coupled versus -uncoupled diffusion models and protein size effects on diffusion coefficient.

values (*Figure 3E*). While 9% of Cy3-SWR1 were slow or immobile in the presence of ATP, 32% were slow or immobile in buffer alone. These results show that while SWR1 does not actively translocate on DNA, binding of ATP increases the mobility of SWR1 on DNA.

## SWR1 and the DNA-binding domain of the Swc2 subunit slide on DNA

SWR1 binding to DNA is mediated in part by the Swc2 subunit, which harbors a positively charged and unstructured DNA-binding domain (DBD) (*Ranjan et al., 2013*). To determine if Swc2 contributes to the diffusive behavior of SWR1 on DNA we compared diffusion coefficients of the SWR1 complex to diffusion coefficients of the DBD of Swc2 (residues 136–345, *Figure 4—figure supplement 1*). We found that Swc2 also diffuses on DNA, however the median diffusion coefficient, $D_{1,obs} = 1.04$ µm$^2$/s ± 0.09, was approximately 40-fold larger than that of SWR1 in the presence of 1 mM ATP (*Figure 4*, Materials and methods). This large difference in measured diffusion coefficients could be due to the difference in size between the small Swc2 DBD and full SWR1 complex or to other DNA-binding components of SWR1 interacting with DNA and increasing friction. Based on theoretical models of rotation-coupled versus -uncoupled diffusion, the scaling relationship between size and diffusion coefficient is consistent with SWR1 and Swc2 DBD utilizing rotationally coupled sliding (*Blainey et al., 2009*; *Figure 4—figure supplement 2*).

Next, we found that both SWR1 and Swc2 DBD show increased diffusion with increasing concentrations of potassium chloride (*Figure 4A*), and each showed decreasing binding lifetimes with increasing salt (*Figure 4B*). Both increased diffusion and decreased binding lifetimes are features of 1D hopping, as the more time a protein spends in microscopic dissociation and reassociation the faster it can move on DNA, but also falls off DNA more frequently (*Bonnet et al., 2008*; *Mirny et al., 2009*). These data are consistent with the single-molecule TIRF data presented earlier (*Figure 1—figure supplement 1E, F*), which also reveals decreased binding lifetimes to DNAs when ionic strength is increased. The TIRF assay also shows that competitor DNA can decrease binding lifetime as would be expected for a protein that hops on DNA and may be prone to alternative binding onto competitor DNA (*Brown et al., 2016*; *Gorman et al., 2007*). We also observed the effects of competitor DNA on Swc2 DBD dwell-time under the same conditions (*Figure 1—figure supplement 1G, H*). Like SWR1, Swc2 DBD experienced shortened dwell-times in the presence of competitor DNA (*Figure 1—figure supplement 1H*). This combined with the observation that high salt decreases the binding lifetime of Swc2 further supports that Swc2 and SWR1 engage in hopping during diffusion. An alternative explanation is that both SWR1 and Swc2 DBD may engage in a so-called 'monkey-bars' mechanism in which binding at a second DNA-binding site promotes dissociation from DNA at the first bound site (*Rudolph et al., 2018*). The Swc2 DBD is likely intrinsically disordered and may be able to bind multiple DNAs simultaneously, as may other DBDs on SWR1 subunits promoting binding onto competitor DNA.

The theoretical upper limit of diffusion for a particle that uses linear translocation (1D hopping) is higher than the theoretical upper limit of diffusion with helically coupled sliding because in the latter there are additional rotational components of friction incurred when circumnavigating the DNA axis (*Blainey et al., 2009*). Based on the molecular weight of SWR1 and Swc2, the theoretical upper limits of 1D diffusion using rotation-coupled versus -uncoupled 1D diffusion can be calculated (Materials and methods). In all conditions measured, the median diffusion of SWR1 is below the upper limit with rotation (*Figure 4C*), consistent with much of the observed diffusion coming from SWR1 engaging in rotationally coupled diffusion. Nonetheless, some individual traces have diffusion coefficients that surpass this theoretical maximum, indicating that there may be alternative modes for engaging with DNA (e.g., infrequent hopping), which allows it to surpass the upper limit with rotation (*Ahmadi et al., 2018*; *Gorman et al., 2010*). A similar phenomenon was observed for Swc2 DBD, which also exhibited median diffusion coefficients below the theoretical maximum with rotation, with some traces having diffusion coefficients above this limit (*Figure 4C*). These trends are consistent with a model in which SWR1 utilizes a majority 1D helically coupled sliding with occasional hopping (or monkey bar-like movements) to diffuse on DNA (*Figure 4D*).

## SWR1 cannot bypass bound dCas9

While the NDR is a region of open chromatin where accessibility to DNA is higher compared to DNA in gene bodies, SWR1 must compete with transcription factors and other DNA-binding proteins for search on this DNA (*Kim et al., 2021*; *Kubik et al., 2019*; *Nguyen et al., 2021*; *Rhee and Pugh*,

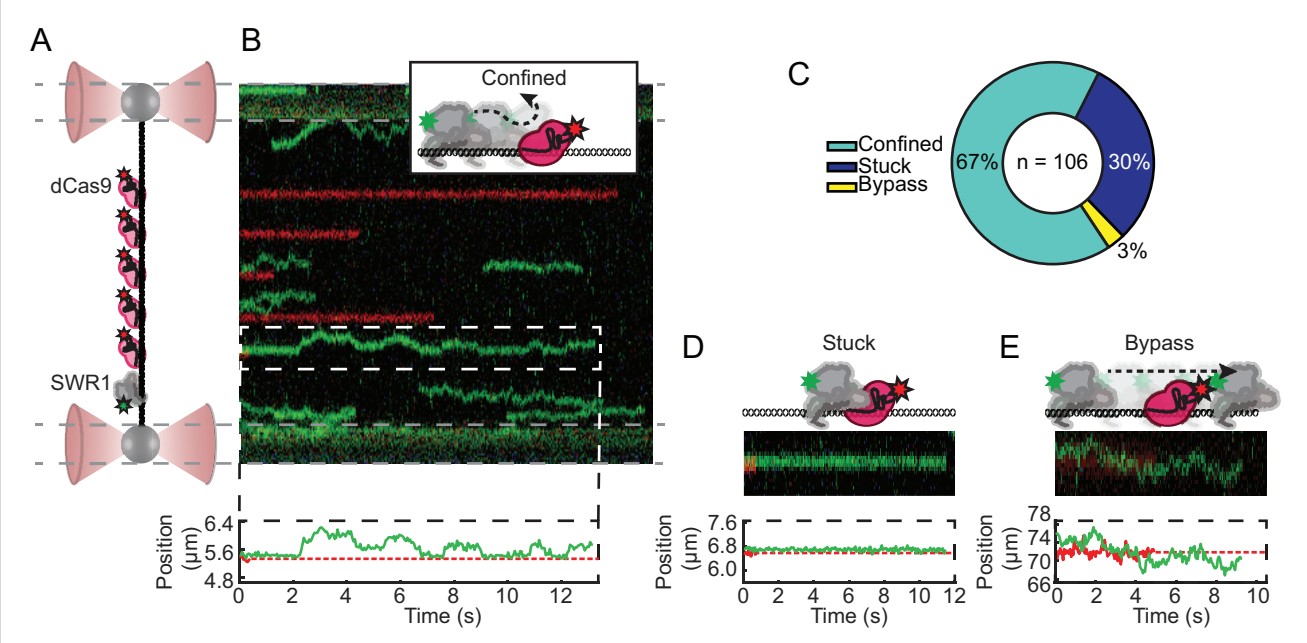

**Figure 5.** dCas9 protein roadblocks confine SWR1 1D diffusion. (**A**) Schematic of the experimental setup: five Cy5-labeled gRNA position dCas9 at five evenly spaced sites along lambda DNA. Note that diffusion is measured in the presence of 1 mM ATP and standard salt conditions (70 mM KCl). (**B**) Example kymograph with five bound dCas9 in red, and an example of a confined diffusion encounter. Schematic, and single-particle tracking trajectory printed above and below. (**C**) Pie chart of the three types of colocalization events with the total number of observations printed therein. (**D**) Example of SWR1 stuck to the dCas9 within limits of detection; schematic, cropped kymograph, and single-particle tracking trajectory shown. Example of a SWR1-dCas9 bypass event; schematic, cropped kymograph, and single-particle tracking trajectory shown. (**B, D, E**) In the example single-particle tracking trajectory, dCas9 is represented as a dashed red line after Cy5 has photobleached, however due to long binding lifetimes of dCas9 we continue to use its position for colocalization analysis.

The online version of this article includes the following source data and figure supplement(s) for figure 5:

**Source data 1.** All colocalization events with classifications indicated.

**Figure supplement 1.** SWR1 bypassing dCas9 was a rare event.

_2012_). Proteins that diffuse on DNA by 1D hopping are sometimes capable of bypassing protein barriers and nucleosomes (_Gorman et al., 2010_; _Hedglin and O'Brien, 2010_). To investigate whether ATP-bound SWR1 can bypass protein barriers, we turned to dCas9, an endonuclease inactive mutant of Cas9, to serve as a programmable barrier to diffusion. We used a dual color single-particle tracking scheme to simultaneously observe Cy3-labeled SWR1 diffusion and the positions of Cy5-labeled dCas9 (_Figure 5_). crRNAs were used to direct dCas9 binding to 5 positions on the lambda DNA using previously validated targeting sequences (_Figure 5A_, _Table 1_, Materials and methods; _Sternberg et al., 2014_). We assume that dCas9 binding far outlasts the photobleaching lifetime of Cy5 (_Singh et al., 2016_), therefore we use the average position of the particle to extend the trace after photobleaching of Cy5 for colocalization analysis. Out of 106 traces with colocalization events, 67% showed SWR1 moving away from dCas9 toward where it came from as if it was reflected from a boundary (_Figure 5B, C_). Another 30% of traces showed SWR1 immobile and colocalized with dCas9 for the duration of the trace, which we describe as stuck (_Figure 5C, D_). Only 3% of all colocalization events exhibited a crossover event (_Figure 5C, E_, _Figure 5—figure supplement 1_). The ability of dCas9 to block SWR1 diffusion in most encounters further supports a model in which SWR1 mainly engages in helically coupled sliding (_Figure 4D_). Infrequent hopping events that colocalize to a dCas9 encounter may contribute to the presence of the rare bypass event (_Figure 5E_, _Figure 5—figure supplement 1_).

## Nucleosomes are barriers to SWR1 diffusion

Diffusion over nucleosomes may also be an important aspect of target search; it is not known whether SWR1 diffusing on an NDR would be confined to this stretch of DNA by flanking +1 and −1 nucleosomes or whether its diffusion could continue into the gene body (_Figure 6A_). To investigate this,

**Table 1.** crRNA sequences for dCas9 binding, and custom oligos sequences for DNA tethering, and dsDNA sequences used in TIRF measurements.

| ID | Identity | Sequence |
|---|---|---|
| 1 | Cas9 crRNA sequence 'lambda 1' | 5'-/ AltR 1/rGrGrC rGrCrA rUrArA rArGrA rUrGrA rGrArC rGrCrG rUrUrU rUrArG rArGrC rU rArU rGrCrU / AltR2/-3' |
| 2 | Cas9 crRNA sequence 'lambda 2' | 5'-/ AltR 1/rGrUrG rArUrA rArGrU rGrGrA rArUrG rCrCrA rUrGrG rUrUrU rUrArG rArGrC rU rArU rGrCrU / AltR2/-3' |
| 3 | Cas9 crRNA sequence 'lambda 3' | 5'-/ AltR 1/rCrUrG rGrUrG rArArC rUrUrC rCrGrA rUrArG rUrGrG rUrUrU rUrArG rArGrC rU rArU rGrCrU / AltR2/-3' |
| 4 | Cas9 crRNA sequence 'lambda 4' | 5'-/AltRl /rCrArG rArUrA rUrArG rCrCrU rGrGrU rGrGrU rUrCrG rUrUrU rUrArG rArGrC rUr ArU rGrCrU / AltR2/-3' |
| 5 | Cas9 crRNA sequence 'lambda 5' | 5'-/AltR 1/rGrGrC rArArU rGrCrC rGrArU rGrGrC rGrArU rArGrG rUrUrU rUrArG rArGrC rUr ArU rGrCrU / AltR2/-3' |
| 6 | 3x-biotin-cos1 oligo | 5'-/5Phos/ AGG TCG CCG CCC TT/iBiodT/TT/iBiodT/TT/3BiodT/-3' |
| 7 | 3x-biotin-cos2 oligo | 5'-/5Phos/ GGG CGG CGA CCT TT/iDigN/TT/iDigN/TT/3DigN/-3' |
| 8 | 20 bp dsDNA | 5'-ttagcaccgggtatctccag-3' |
| 9 | 40 bp dsdna | 5'-ttagcaccgggtatctccagatcgatgcaagggcgaattc-3' |
| 10 | 60 bp dsdna | 5'-ttagcaccgggtatctccagatcgatgcaagggcgaattctgcagatatccatcacactg-3' |
| 11 | 80 bp dsdna | 5'-ttagcaccgggtatctccagatcgatgcaagggcgaattctg cagatatccatcacactggcggccgctcgagcatgcat-3' |
| 12 | 100 bp dsdna | 5'-ttagcaccgggtatctccagatcgatgcaagggc gaattctgcagatatccatcacactggcggccgctcgagcatgcatctagagggcccaattcgccc-3' |
| 13 | 150 bp dsdna | 5'-ttagcaccgggtatctccagatcgatgcaagggcgaattctgcagatatccatcacact ggcggccgctcgagcatgcatctagagggcccaattcgccctatagtga gtcgtattacaattcactggccgtcgttttacaacgtcgtga-3' |

we monitored SWR1 diffusion on sparse nucleosome arrays reconstituted on lambda DNA. Nucleosomes were formed at random sites along lambda DNA using salt gradient dialysis, as has been done previously (*Gruszka et al., 2020*; *Visnapuu and Greene, 2009*; *Figure 6—figure supplement 1A*, Materials and methods). On average, 40 ± 5 nucleosomes were incorporated onto the lambda nucleosome arrays as shown by nucleosome unwrapping force–distance curves (*Figure 6B–E*) nucleosomes showed detectable unwrapping at forces 15 pN or greater (*Figure 6B*; *Brower-Toland et al., 2002*; *Fierz and Poirier, 2019*), with a characteristic lengthening of the array by ~25 nm with each unwrapping event (*Figure 6C*; *Spakman et al., 2020*). To determine a compaction ratio which could be used to estimate the number of nucleosomes per array in the case where the array breaks before it has been fully unwrapped, unwrapping events were counted and related to the total length of the array at 5 pN (*Figure 6—figure supplement 1B*, Materials and methods).

Overall, the behavior of SWR1 on lambda nucleosome arrays was notably different than on naked lambda DNA (*Figure 6F, G*, *Figure 6—figure supplement 1C–E*). The mean MSD for SWR1 on naked DNA increases linearly at short time scales (<2 s), whereas the mean MSD for SWR1 on the lambda nucleosome array plateaus over this same time scale, indicative of confined 1D diffusion (*Figure 6F*). The degree to which diffusion is confined can be described by $\alpha < 1$ where $MSD = Dt^{\alpha}$. Whereas SWR1 on naked DNA has an $\alpha = 0.88$ over a 2-s time scale, SWR1 on the lambda array has an $\alpha = 0.24$ reflecting considerable confinement. By fitting the MSD curve to an exponential function, the mean MSD appears to approach a limit of 0.054 μm² (*Figure 6—figure supplement 1C*). Assuming an even distribution of an average of 40 nucleosomes per array (*Figure 6E*), the mean distance between nucleosomes is equal to 0.35 μm; whereas the length of DNA to which SWR1 diffusion is confined is approximately 0.23 μm, determined from the square root of the MSD limit described above. Representative traces show signs of confinement, as more immobile segments dominate the trace and the range of SWR1 exploration becomes confined (*Figure 6G*). Moreover, diffusion coefficients for traces on the nucleosome array are overall smaller, despite the presence of 1 mM ATP (*Figure 6—figure supplement 1D*). The percentage of traces exhibiting zero to low diffusion is increased in this condition, which is further seen when overlaying all trajectories of SWR1 diffusion in the absence versus

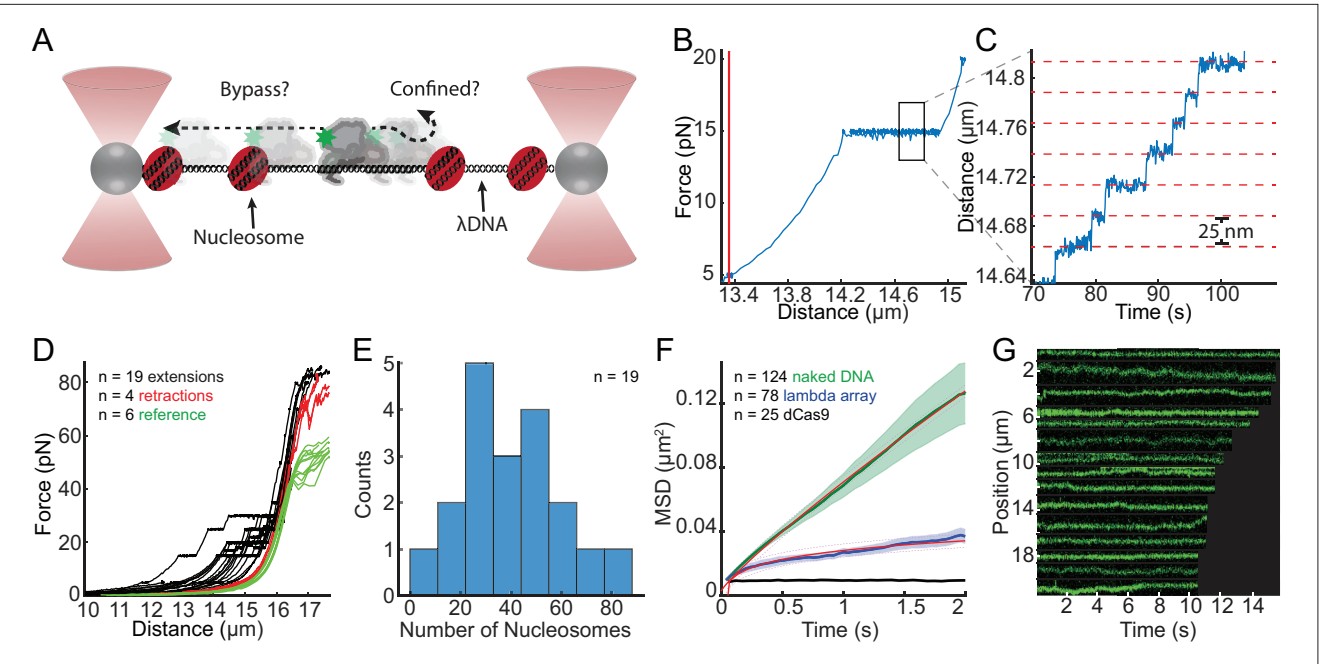

**Figure 6.** Nucleosomes confine SWR1 1D diffusion. (**A**) Schematic of the experimental setup, with experimental question depicted therein. Note that diffusion is measured in the presence of 1 mM ATP and standard salt conditions (70 mM KCl). (**B**) Example force–distance curve showing that at 15 pN nucleosomes begin to unwrap. Vertical red line shows the length of the nucleosome array 5 pN. (**C**) Example unwrapping events that result in characteristic lengthening of 25 nm at this force regime. (**D**) Lambda nucleosome arrays extension (unwrapping) and retraction curves, with a reference naked DNA force–distance curves. Black curves are unwrapping curves where the force is clamped at either 20, 25, or 30 pN to visualize individual unwrapping events; red curves are the collapse of the DNA after unwrapping nucleosomes; green curves are reference force extension plots of lambda DNA without nucleosomes. (**E**) Histogram of the number of nucleosomes per array determined from the length of the array at 5 pN and the compaction ratio. (**F**) Mean squared displacements (MSDs) are fit over the first 2 s to MSD = $Dt^{\alpha}$, the red lines represent the fits with 95% confidence interval shown as dashed lines. SWR1 diffusing on naked DNA (green curve), on lambda nucleosome arrays (blue), and for comparison dCas9 (black). (**G**) Representative SWR1 particles diffusing on the nucleosome arrays are cropped and arranged by the length of the trace.

The online version of this article includes the following source data and figure supplement(s) for figure 6:

**Source data 1.** Data underlying panels B, C, and E.

**Figure supplement 1.** Primary lambda nucleosome array characterization.

**Figure supplement 1—source data 1.** Gel images (Coomassie and Cy3 scans).

**Figure supplement 1—source data 2.** Data underlying panels B and D.

**Figure supplement 2.** Instantaneous diffusion analysis of SWR1 diffusion on the lambda nucleosome array.

the presence of nucleosomes (***Figure 6—figure supplement 1E***). These data, therefore, suggest that SWR1 diffusion is confined to the space between nucleosomes.

Instantaneous diffusion analysis of SWR1 bound to nucleosome arrays suggests that SWR1 may be transiently engaging the nucleosomes in the context of a 1D diffusion-mediated encounter. For this analysis, diffusion coefficients are calculated for all windows of 0.4 s in length, allowing for an understanding of how diffusivity changes over the length of a trajectory (***Figure 6—figure supplement 2***). Aiding in the interpretation of overall trends, diffusion is assigned to one of three categories: fast diffusion ($D_{coef} > 0.04$ μm²/s), slow/immobile diffusion ($D_{coef} < 0.01$ μm²/s), or medium diffusion (between the threshold values for fast and slow diffusion) (***Figure 6—figure supplement 2A***). Slow and immobile are technically not distinguishable in this study as higher instrumental spatial and temporal resolution would be required. If SWR1 is engaging the nucleosome upon encounter, one may predict that SWR1 diffusion on the sparse nucleosome array would begin in high or medium diffusion and transition into a slow/immobile state.

In this study, a majority of DNA-binding events by SWR1 occurred before imaging was initiated; when averaging across the length all trajectories from the start of observation, SWR1 diffusion on the sparse nucleosome array does not decrease or exhibit a higher immobile fraction (***Figure 6—figure***

*supplement 2B*). While this imaging setup limited the extent to which instantaneous diffusion analysis could be used to observe the initial nucleosome capture event, we did observe that the overall percentages within individual trajectories spent in the immobile state is drastically increased for nucleosome arrays, indicating a longer duration for the nucleosome-captured state (*Figure 6—figure supplement 2C*). Furthermore, a dwell-time analysis of the SWR1 immobile state within trajectories (*Figure 6—figure supplement 2D–F*) shows that SWR1 spends more time immobile on the nucleosome array as compared to on naked DNA (*Figure 6—figure supplement 2F*). In summary, SWR1 diffusion on DNA is confined by nucleosomes and may result in transient nucleosomal interactions.

## Discussion
### Reducing the dimensionality of nucleosome target search

Our single-molecule tracking data show that SWR1 slides on DNA, which is a novel finding for a chromatin remodeler. Moreover, SWR1 scans DNA with a diffusion coefficient comparable to other well-characterized proteins that utilize facilitated diffusion to bind specific DNA sequences or lesions (*Ahmadi et al., 2018*; *Blainey et al., 2006*; *Gorman et al., 2010*; *Kamagata et al., 2020*; *Porecha and Stivers, 2008*; *Tafvizi et al., 2011*; *Tafvizi et al., 2008*; *Vestergaard et al., 2018*). Without 1D sliding, the search process of SWR1 for its target nucleosome would be dependent solely on 3D collisions with nucleosomes. In the yeast genome, there are approximately 61,568 annotated nucleosomes (*Jiang and Pugh, 2009*; *Kubik et al., 2015*), of which 4576 are identified as potential +1 nucleosomes enriched in H2A.Z (*Tramantano et al., 2016*). Since only 7% of nucleosomes are targets of SWR1 histone exchange, we believe that +1 nucleosomes use their adjacent NDRs as antennas, promoting SWR1 binding and 1D search to encounter flanking nucleosomes (*Mirny et al., 2009*). This increased efficiency in target localization through dimensional reduction of the search process may be one that could extend to other chromatin remodelers that act on nucleosomes adjacent to the NDR, such as RSC, SWI/SNF, ISW2, and INO80 (*Kim et al., 2021*).

While this study establishes for the first time that the chromatin remodeler SWR1 can diffuse on DNA, and that such diffusion is confined by dCas9 and nucleosomes, many questions remain. First, while SWR1 may be guided to its main target nucleosome, +1, via binding to the NDR, how SWR1 preferentially recognizes the +1 nucleosome over the −1 nucleosome on the other side of the NDR remains unanswered. While there is evidence that acetylation of the +1 nucleosome may help establish the directionality of the interaction when assessing the two NDR-flanking nucleosomes (*Ranjan et al., 2013*; *Zhang et al., 2005*), data from our study indicate that even nucleosomes lacking acetylation can reduce the diffusion of the SWR1 complex possibly through nucleosome engagement. Therefore, it will be necessary to measure the fold difference in dwell-time on nucleosomes, in the context of 1D diffusion, that do and do not contain acetylation marks in order to determine the magnitude of the effect that the 1D search process would have on directing binding to the +1 to −1 nucleosome. Of note, there are times where both NDR-flanking nucleosomes will be substrates of SWR1 and are equally enriched for H2A.Z incorporation, as is the case when both nucleosomes are the +1 nucleosomes of divergently transcribed genes (*Bagchi et al., 2020*). Additionally, H2A.Z is preferentially enriched at inactive promoters (*Li et al., 2005*), this phenomenon should be revisited in the light of 1D diffusion as it is possible that regulatory mechanisms that affect transcription of these genes also impact the ease with which SWR1 can perform 1D diffusion.

Furthermore, the DNA used as a substrate in this study, lambda DNA, is different in mechanical properties, owing to differences in sequence composition, compared to yeast promoters and may therefore not be the most biologically representative substrate for studying 1D diffusion of SWR1. A sequencing study discovering the intrinsic flexibility of different DNA sequences helped establish that promoter NDR regions from yeast are highly rigid and this unique property compared to gene-body DNA helps instruct the chromatin remodeler, INO80, a close relative of SWR1, to position nucleosomes at the boundary of the rigid NDR helping to establish the defined positioning of the +1 nucleosome (*Basu et al., 2021*). It is possible that sensing changes in DNA rigidity while performing a 1D search could allow SWR1 to distinguish +1 from −1 nucleosomes. Finally, in our study, a distinction is not made between NDRs and nucleosome-free regions (NFRs) in contemplating the relevance of a 1D search mechanism in targeting SWR1 to the +1 nucleosome. The distinction between these two types of promoter architectures was recently further developed via the implementation of ChIP-exo/

seq to comprehensively discover occupancies of fragile nucleosome in addition to bound transcription factors at yeast promoters (*Rossi et al., 2021*). The complexity of the integrated transcriptional regulatory networks and 3D chromatin architecture is a caveat to the simplified model presented in this study and may limit the magnitude to which a 1D search process factors into target nucleosome-binding in vivo. Fragile nucleosomes in NDRs effectively shorten the length of nucleosome-free linker DNA proximal to the +1 nucleosome, diminishing the kinetic binding preference for that NDR. Fragile nucleosomes, in addition to transcription factors, may also impede diffusion, increasing the time scale required to encounter flanking nucleosomes via 1D diffusion. In vivo binding of SWR1, as shown via ChIP-seq, has already been shown to be reduced by fragile nucleosome occupancy of NDRs (*Ranjan et al., 2013*; *Venters and Pugh, 2009*). Therefore, while the observations presented in this study are an exciting launching point into studies of chromatin remodeler 1D diffusion, future work with additional promoter-binding proteins is needed to assess the relevance of these biophysical measurements to in vivo target binding.

## ATP binding facilitates SWR1 target search and diffusion on DNA

We observed that SWR1 diffusion is increased in the presence of ATP, and that substitution with ATPγS also results in similar increased diffusion suggesting that this enhancement is mediated by nucleotide binding rather than hydrolysis. SWR1 requires ATP to perform the histone exchange reaction, and basal levels of ATP hydrolysis when any one of the required substrates for the histone exchange reaction is missing is low (*Luk et al., 2010*). This includes the scenario where SWR1 is bound to DNA in the absence of the nucleosome and H2A.Z/H2B dimer. Therefore, we do not expect SWR1 diffusion in the presence of 1 mM ATP to be modulated by ATP hydrolysis, which is consistent with our findings. Binding of nucleotide cofactor has been shown to produce conformational changes in ATPases that can affect their diffusion on DNA (*Gorman et al., 2007*). The core ATPase domain of SWR1, Swr1, like other chromatin remodelers, belongs to the SF2 of translocases which have two lobes that switch between an open and closed conformation with ATP binding and hydrolysis (*Beyer et al., 2013*; *Nodelman et al., 2020*). It is therefore possible that the ATP bound closed conformation of the core ATPase results in a DNA-binding interface, distributed across accessory domains, that is more conducive to diffusion on DNA, contributing to the enhanced diffusion of SWR1 in the presence of ATP or ATPγS. In the present study, we further investigated a DNA-binding subunit, Swc2, which forms an extended interface with the core ATPase (*Willhoft et al., 2018*). In addition to the changes in the contacts that the translocase domain makes with DNA in the closed versus open form, it is possible that ATP modulates how Swc2 engages with the DNA through conformational changes propagated from Swr1. Swc2 appears to be an important accessory subunit for 1D diffusion, as we were able to show that in isolation, the DBD of Swc2 diffuses on DNA. We show that, like SWR1, Swc2 likely utilizing a combination of sliding and hopping to diffuse on DNA, while exhibiting a much-increased diffusion coefficient owning to its smaller size. Future studies will aim to address how ATP binding to the catalytic subunit may alter its conformation and/or that of Swc2 (or another DNA-binding component) leading to enhanced diffusion of the SWR1 complex.

Conformations that result in slower sliding presumably become trapped in free energy minima along the DNA where the DNA sequence or the presence of DNA lesions results in a more stably bound DNA–protein interaction (*Gorman et al., 2007*). While it remains unknown whether SWR1 interacts with different sequences of DNA differently in the context of sliding, we believe this may be a possibility since we observe a distribution in diffusion coefficients within any single condition which would not be expected if the energetic costs of binding substrate were equal everywhere. The NDR is rich in AT-content; therefore, one might imagine that SWR1 may have evolved to be better at scanning DNA with high AT-content (*Chereji et al., 2018*). Lambda DNA, the DNA substrate used in this study, has asymmetric AT-content, which has been shown to affect nucleosome positioning during random deposition (*Visnapuu and Greene, 2009*). Future studies of chromatin remodeler 1D diffusion are needed to address this possibility.

## SWR1 and Swc2 predominantly slide with diffusion confined between roadblocks

The way a protein engages with DNA during 1D search can have impacts on both scanning speed and target localization. For instance, a protein that maintains continuous contact with the DNA in

part through charge–charge interactions with the phosphate backbone will predominantly utilize helically coupled sliding. By contrast, a protein that dissociates just far enough from the DNA for cation condensation on the phosphate backbone to occur before quickly reassociating will utilize linear hopping to perform short 3D searches before reassociating at a nearby site on the DNA (*Mirny et al., 2009*). Proteins that hop on DNA therefore have increased diffusion with increased monovalent cation concentration, as a higher screening potential results in more frequent hops. SWR1 and the DBD of the Swc2 subunit both become more diffusive as the concentration of potassium chloride is increased (*Figure 4A*), which indicates that both utilize some degree of hopping when diffusing on DNA.

Nonetheless, the observed diffusion for both SWR1 and Swc2, on average, falls within a range expected for a protein that predominantly uses a sliding mechanism to diffuse on DNA. In order for a protein to slide or hop on DNA, the energy barrier ($\Delta G^{\ddagger}$) to break the static interaction and dynamically engage with the DNA following the parameters of either the sliding or hopping model must be less than $\approx 2\ k_\mathrm{B}T$ (*Ahmadi et al., 2018*; *Gorman et al., 2007*; *Slutsky and Mirny, 2004*). Based on the molecular weight of SWR1 and Swc2, the upper limit of 1D diffusion was estimated for both the sliding and hopping model (*Figure 4C*, Materials and methods). The upper limit of diffusion coefficients for rotation-coupled sliding-only diffusion is lower than hopping-only diffusion due to the rotational component increasing friction in the sliding model. We found that most particles for either SWR1 or Swc2 fall below the estimated upper limit for sliding diffusion. This observation indicates that, averaged over the length of the trace, the energetic barrier to exclusively hop along DNA is too large, whereas the energy barrier for sliding diffusion is permissive ($<2\ k_\mathrm{B}T$). Therefore, while both SWR1 and Swc2 DBD can engage in hopping, both on average utilize sliding diffusion as exhibited by their slow diffusion.

Sliding as a predominant component of the SWR1 interaction with DNA is further evidenced by the observation that SWR1 can neither bypass a dCas9 protein roadblock nor nucleosomes with high efficiency. Other studies have found that proteins that utilize sliding as the predominant form of 1D diffusion cannot bypass proteins or nucleosomes (*Brown et al., 2016*; *Gorman et al., 2010*; *Hedglin and O'Brien, 2010*), whereas a protein that predominantly hops may be able to bypass these obstacles. The utilization of hopping diffusion has been described as a trade-off between scanning speed and accuracy, with proven implications in target sequence bypass by the transcription factor LacI (*Marklund et al., 2020*). Whether the same may be true for chromatin remodelers in search of specific nucleosomes is yet to be reported.

## Concluding remarks

Single-particle tracking in vivo has shown that approximately 47% of SWR1 molecules are bound to chromatin and the remainder is performing 3D diffusion (*Ranjan et al., 2020*). Once bound (e.g., near the center of an average NDR of ~150 bp) our findings suggest that SWR1 would require 46 ms (see Materials and methods) to scan and encounter a flanking nucleosome by 1D diffusion at 0.024 µm$^2$/s. While our study lacks cellular data demonstrating the essentiality of 1D scanning for SWR1 target localization efficiency and SWR1 function in vivo, SWR1 bound to chromatin has been shown to have a bound-state diffusion coefficient that is ~twofold larger than the diffusion of chromatin itself (SWR1 D bound 0.063 ± 0.0003 µm$^2$/s versus histone H2B D bound 0.028 ± 0.0002 µm$^2$/s) (*Ranjan et al., 2020*). A later study found that other yeast chromatin remodelers that act on nucleosomes adjacent to the NDR (RSC, SWI/SNF, ISW2, and INO80) are similarly more diffusive than histone H2B in their chromatin-bound states, and are, similar to SWR1, dependent on the ability of their ATPase domains to bind or hydrolyze ATP in order to exhibit this enhanced bound diffusion (*Kim et al., 2021*). While several factors could result in this enhanced bound diffusion, such as enhanced diffusion of promoter chromatin or intersegmental transfer of SWR1 (a topic not explored in our study), 1D diffusion may be an important contributor to this enhanced chromatin-bound diffusion. A single-molecule TIRF study on the molecular mechanism of the SWR1 histone exchange reaction recently reported that when complexed with a canonical nucleosome and the H2A.Z–H2B dimer, SWR1 can rapidly perform the ATP hydrolysis-dependent histone exchange reaction on average in 2.4 s (*Poyton et al., 2022*). The SWR1-catalyzed histone H2A.Z exchange on chromatin may therefore be an intrinsically rapid event that occurs on a time scale of seconds. While 1D diffusion should in principle allow SWR1 to encounter either the +1 or −1 nucleosome at the ends of an NDR, we anticipate that future studies of diffusion toward acetylated nucleosomes may provide insights on how the complex is preferentially enriched

**Table 2.** Protein construct sequence.

| Identity | Sequence |
| --- | --- |
| Swc2 DNA-binding domain (italicized) with site of cysteine insertion in bold | HHHHHHSSGLEVLFQGPH**C**IRRQELLSRKKRNK RLQKGPVVIKKQKPKPKSGEAIPRSHHTHEQLN AETLLLNTRRTSKRSSVMENTMKVYEKLSKAEK KRKIIQERIRKHKEQESQHMLTQEERLRIAKETE KLNILSLDKFKEQEVWKKENRLALQKRQKQKF QPNETILQFLSTAWLMTPAMELEDRKYWQEQ LNKRDKKKKKYPRKPKKNLNLGKQDASDDKKRE |

at the +1 nucleosome that bears a higher level of histone acetylation. Future investigations should consider analysis of 1D diffusion on nucleosome arrays that mimic the natural nucleosome arrangement and histone modifications of NDRs and gene bodies and will provide important biophysical and temporal insights on how SWR1 undergoes target search to capture its nucleosome substrates at gene promoters. They might also include an investigation of mutations to SWR1 subunits that permit histone exchange but abrogate diffusivity on DNA, which would establish the importance of this biophysical property to SWR1 activity in vivo. Extension of this approach to other ATP-dependent chromatin remodelers and histone modification enzymes will facilitate understanding of the cooperating and competing processes on chromatin resulting in permissive or nonpermissive architectures for eukaryotic transcription.

# Materials and methods
## Protein purification, fluorescence labeling, and functional validation (SWR1 and Swc2)

The SWR1 complex labeled only on Swc7 was constructed as has been previously documented (*Poyton et al., 2022*). We demonstrated that the fluorescently labeled SWR1 complex maintains full histone exchange activity (*Figure 1—figure supplement 1B*). For this assay, 1 nM SWR1, 5 nM nucleosome, and 15 nM ZB-3X flag were combined in standard SWR1 reaction buffer (25 mM HEPES (4-(2-hydroxyethyl)-1-piperazineethanesulfonic acid) pH 7.6, 0.37 mM ethylenediaminetetraacetic acid (EDTA), 5% glycerol, 0.017% NP40, 70 mM KCl, 3.6 mM MgCl$_2$, 0.1 mg/ml bovine serum albumin [BSA], 1 mM 2-mercaptoethanol [BME]) supplemented with 1 mM ATP, and the reaction was allowed to proceed for 1 hr before being quenched with (100 ng) lambda DNA. The product was run on a 6% native mini-PAGE run in 0.5× Tris/Borate/EDTA (TBE) buffer as has been previously reported (*Ranjan et al., 2013*).

The DBD of Swc2 (residues 136–345) was cloned into a 6× his-tag expression vector with a single cysteine placed directly before the N-terminus of the protein for labeling purposes (*Table 2*). The Swc2 DBD was purified after expression under denaturing conditions using Ni-NTA affinity purification. After purification, the Swc2 DBD was specifically labeled in a 30-fold excess of Cy3-maleimide. After fluorophore labeling, the Swc2 DBD was Ni-NTA purified a second time to remove any excess free dye. The product was then dialyzed overnight at 4°C into refolding buffer 20 mM Tris pH 8.0, 0.5 M NaCl, 10% Glycerol, 2 mM β-mercaptoethanol, 0.02% NP40, and 1 mM PMSF (phenylmethylsulfonyl fluoride) as has been previously documented (*Ranjan et al., 2013*). Pure protein was stored as aliquots at −80°C until time of use. Sodium dodecyl sulfate–polyacrylamide gel electrophoresis reveals a pure Cy3-labeled product (*Figure 4—figure supplement 1*).

## dCas9 crRNAs, fluorescent tracrRNA annealing, and RNP assembly

dCas9 was purchased from Integrated DNA Technologies (IDT), as Alt-R S.p.d Cas9 Protein V3 and stored at −80°C until Ribonucleoprotein (RNP) assembly. crRNAs used to target 5 sites along lambda DNA were ordered from IDT. The crRNAs used were previously validated (*Sternberg et al., 2014*) and are listed in *Table 1* (1–5). Custom 3′-amine modified tracrRNA was ordered from IDT and reacted with monoreactive NHS-ester Cy5 dye (Fisher Scientific cat# 45-001-190). The labeled product was reverse-phase HPLC purified. crRNA and Cy5-tracrRNA were annealed in IDT duplex buffer (cat# 11-01-03-01) in equimolar amounts by heating the mixture to 95°C for 5 min and allowing it to cool

to room temperature slowly on the benchtop. RNP complexes were assembled by mixing annealed guide RNA and dCas9 in a 1.5:1 molar ratio and allowing the mixture to stand at room temperature for 15 min prior to use. Aliquoted RNPs were flash frozen and stored at −80°C until time of use. Buffers for RNP assembly and cryo-storage are the same and contains: 20 mM Tris–HCl pH 7.5, 200 mM KCl, 5% glycerol, and 1 mM TCEP (tris(2-carboxyethyl)phosphine). dCas9 RNPs were diluted to 10 nM just prior to imaging in 1× New England Biolabs (NEB) 3.1 (cat# B7203S).

## Lambda DNA preparation

Biotinylated lambda DNA used in SWR1 sliding on naked DNA assays was purchased from LUMICKS (SKU: 00001). Lambda DNA used in nucleosome array assays was made with 3 biotins on one end, and 3 digoxigenin on the other end using the following protocol. Custom oligos were ordered from IDT with sequences listed in *Table 1* (6–7). Lambda DNA was ordered from NEB (cat# N3011S). Oligo 1 was annealed to lambda DNA by adding a 25-fold molar excess of oligo to lambda DNA, in an annealing buffer containing 30 mM HEPES pH 7.5 and 100 mM KCl. This mixture was heated to 70°C for 10 min and allowed to cool slowly to room temperature on the benchtop. 2 µl of NEB T4 DNA ligase (400 U, cat# M0202S) was added along with T4 DNA ligase buffer containing ATP and allowed to incubate at room temperature for 30 min. Then 50-fold molar excess of oligo 2 was added to the mixture along with an additional 1 µl of T4 DNA ligase and T4 DNA ligase buffer (NEB) with ATP adjusting for the change in volume and allowed to incubate at room temperature for 30 min. The resulting mixture was heat inactivated at 65°C for 10 min. End-labeled lambda DNA was purified using Qiaex II gel-extraction DNA clean-up kit following the manufactures' instructions (Qiagen cat# 20021).

## Lambda nucleosome array construction and validation

A salt gradient dialysis approach was used to reconstitute nucleosomes onto lambda DNA using methods optimized in the lab based on previously established protocols (*Luger et al., 1999*; *Vary et al., 2003*). Buffers used in this reconstitution are as follows: high-salt buffer (10 mM Tris–HCl pH 7.5, 1 mM EDTA pH 8, 2 M NaCl, 0.02% NP-40, 5 mM BME) and low-salt buffer (10 mM Tris–HCl pH 7.5, 1 mM EDTA pH 8, 50 mM NaCl, 0.02% NP-40, 5 mM BME). Cy5-labeled H3 containing octamer, with the same composition and preparation as previously used (*Ranjan et al., 2013*), was titrated onto the lambda DNA in the follow molar ratio to DNA (10:1, 50:1, 100:1, 200:1, 500:1, 700:1). Reconstitution reactions were prepared in 10 mM Tris pH 7.5, 1 mM EDTA pH 8, 0.1 mg/ml BSA Roche (cat # 10711454001), 5 mM BME. Any dilutions of octamer were prepared in octamer refolding buffer (10 mM Tris–HCl pH 7.5, 1 mM EDTA pH 8, 2 M NaCl, 5 mM BME). A 16-hr dialysis was set up by placing the reconstitution mixture in a 7-kDa MWCO Slide-A-Lyzer MINI Dialysis Device (Thermo Scientific cat # 69560) and placed in a flotation device in high-salt buffer. Low-salt buffer was slowly dripped into high-salt buffer for the duration of the dialysis with constant stirring. At the end of this dialysis period, the dialysis solution was dumped and replaced by 100% low-salt buffer and allowed to dialyze for an additional hour. The reconstitution efficiency was first assessed using an EMSA (*Figure 6—figure supplement 1A*). Lambda nucleosome arrays were loaded on a 0.5% agarose gel made with Invitrogen UltraPure Agarose (Fisher Scientific cat # 16-500-500) and 0.25× TBE. Sucrose loading buffer without added dyes was used to load samples on the gel. The gel was run for 1 hr and 45 min at 100 V in 0.25× TBE.

Arrays contained a variable number of nucleosomes, where the mean number of nucleosomes per array is 40 ± 5 (standard deviation) for a total of 19 arrays. The number of nucleosomes per array was estimated from the length of the lambda nucleosome array at 5 pN force before and after nucleosome unwrapping. On average, approximately 34.6 nm of lengthening at 5 pN corresponded to the unwrapping of a single nucleosome, therefore the difference in length before and after unwrapping was used to estimate the number of nucleosomes per array.

## Dual optical tweezers and confocal microscope setup and experimental workflow

The LUMICKS cTrap (series G2) was used for optical tweezer experiments, configured with two optical traps. The confocal imaging laser lines used were 532 nm (green) and 640 nm (red) in combination with emission bandpass filters 545–620 nm (green) and 650–750 nm (red). A C1 type LUMICKS microfluidics chip was used. The microfluidics system was passivated at the start of each day of imaging

as follows: 0.1% BSA was flowed at 0.4 bar pressure for 30 min, followed by a 10 min rinse with PBS at 0.4 bar pressure, followed by 0.5% Pluronic F-127 flowed at 0.4 bar pressure for 30 min, followed by 30 min rinse with PBS at 0.4 bar pressure. For SWR1 sliding on naked DNA, 4.2 μm polystyrene beads coated in streptavidin (Spherotech cat# SVP-40-5) were caught in each trap, and LUMICKS biotinylated lambda DNA was tethered. Both traps had trap stiffness of about 0.8 pN/nm. For SWR1 sliding on lambda nucleosome array, a 4.2-μm polystyrene bead coated in streptavidin was caught in trap 1, and a 2.12-μm polystyrene bead coated in anti-digoxigenin antibody (Spherotech cat# DIGP-20-2) was caught in trap 2 which is upstream in the path of buffer flow to trap 1. For this configuration, trap 1 had a trap stiffness of about 0.3 pN/nm whereas trap 2 had a trap stiffness of about 1.2 pN/nm. The presence of a single tether was confirmed by fitting a force extension plot to a worm like chain model in real time while collecting data using LUMICKS BlueLake software. For confocal scanning, 1.8 μW of green and red laser power were used. For most traces, the frame rate for SWR1 imaging was 50 ms, whereas for Swc2 it was 20 ms. Experiments were performed at room temperature. SWR1 and Swc2 were both imaged in histone exchange reaction buffer (25 mM HEPES pH 7.6, 0.37 mM EDTA, 5% glycerol, 0.017% NP40, 70 mM KCl, 3.6 mM MgCl$_2$, 0.1 mg/ml BSA, 1 mM BME) made in imaging buffer. dCas9 was added to the flow chamber in Cas9-binding buffer (20 mM Tris–HCl pH 8, 100 mM KCl, 5 mM MgCl$_2$, 5% glycerol) made in imaging buffer. Imaging buffer (saturated Trolox [Millipore Sigma cat# 238813], 0.4% dextrose) is used in place of water when preparing buffers. All buffers were filter sterilized with a 0.2 μm filter prior to use.

## TIRF-based binding kinetics assay and analysis

We colocalized SWR1 and Swc2 DBD binding to Cy5-labeled dsDNAs of different lengths for real-time binding kinetic measurements (*Figure 1—figure supplement 1D–H*). These experiments were all conducted using flow cells made with PEG-passivated quartz slides using previously detailed methods (*Roy et al., 2008*). The appropriate biotinylated Cy5-labeled DNA was immobilized on the surface of the PEG-passivated quartz slide using neutravidin. After DNA immobilization, the channels of the flow cell were washed to remove free DNA and imaging buffer was flowed into the channel. Next, 5 nM Cy5-SWR1 in imaging buffer was flowed into the channel immediately after starting image acquisition. A standard smFRET imaging buffer with oxygen scavenging system was used as has been previously established (*Joo and Ha, 2012*). The first 10 frames (1 s) of each imaging experiment were collected using Cy5 excitation so that all Cy5-DNA spots could be identified. The remaining 299 s of the movie were collected under Cy3 excitation so that Cy3-SWR1 could be imaged. Data analysis was carried out using homemade IDL scripts for image analysis and MATLAB scripts for data analysis. The data were analyzed so that all the Cy5-DNA molecules in an image were identified from the first second of the movie under Cy5 excitation. Next, the Cy3 intensity was monitored for the remainder of the movie for each DNA molecule. SWR1 binding to DNA was detected by a sharp increase in Cy3 signal in spots that had Cy5 signal.

The on-rate ($k_{bind}$) was defined as the time between when Cy3-SWR1 was injected into the imaging chamber to when Cy3-SWR1 first bound to a specific DNA molecule resulting in an increase in Cy3 intensity. The off-rate ($t_{bound}$) was defined as the length of time Cy3-SWR1 was bound to a DNA molecule. While only one on-rate measurement could be conducted for one DNA molecule, multiple off-rate measurements could be made as one DNA molecule was subjected to multiple Cy3-SWR1-binding events. Binding events where more than one SWR1 were bound to the DNA were excluded from the off-rate analysis. Off-rate measurements under different laser intensities were made by measuring the laser power immediately prior to the imaging experiment (*Figure 1—figure supplement 1D*). All experiments were conducted using imaging channels from the same quartz slide to minimize differences in laser intensity that can result from changes in shape of the TIRF spot. All sequences of DNAs used for DNA length-dependent binding and unbinding measurements are provided in *Table 1* (8–13).

## Single-particle tracking and data analysis

LUMICKS Bluelake HDF5 data files were initially processed using the commercial Pylake Python package to extract kymograph pixel intensities along with corresponding metadata. Particle tracking was then performed in MATLAB (MathWorks). First, spatially well-separated particles were individually segmented from full-length kymographs containing multiple diffusing particles. Next, for each time

step, a one-dimensional Gaussian was fit to the pixel intensities to extract the centroid position of the particle in time. Then the MSD for each time lag was calculated using:

$$\text{MSD}\left(n, N\right) = \sum_{i=1}^{N-n} \frac{\left(X_{i+n} - X_i\right)^2}{N-n} \tag{1}$$

where $N$ is the total number of frames in the trace, $n$ is the size of the time lag over which the MSD is calculated, $i$ is the sliding widow over which displacement is measured, and $X$ is the position of the particle. Since particles exhibit Brownian diffusion, the diffusion coefficient for each particle was then calculated from a linear fit to the initial portion of the MSD versus time lag plot by solving for $D$ using: $MSD = 2Dt$. For mean MSD plots, traces with the same frame rate were averaged together, resulting in a slightly different $n$ value as compared to all trajectories in a condition.

For the linear fit, the number of points included varied to optimize for a maximal number of points fit with the highest Pearson correlation ($r^2$) and a p value lower than 0.05. For particles where this initial best fit could not be found, the first 25% of the trace was linearly fit. Fits that produced negative slope values corresponded to traces where particles are immobile; to reflect this, negative slopes were given a slope of 0. Finally, outlier traces with diffusion coefficients greater than 0.14 µm²/s for SWR1 or 5 µm²/s for Swc2 were dropped; in every case this consisted of less than 3% of all traces. The distribution of diffusion coefficients estimated using this method was almost identical to what is produced using an alternative method which extracts diffusion coefficients using a linear fit from time lags 3–10 rejecting fits with $r^2 < 0.9$ (*Tafvizi et al., 2008*) (data not shown). A summary of statistics as well as criteria for excluding traces is provided in *Table 3*. Also included are the number of biological and technical replicates per condition. A biological replicate is defined as a fresh aliquot of protein imaged on a different imaging day, whereas a technical replicate is the number of distinct DNAs or nucleosome arrays used per imaging condition; a single DNA could accommodate one or more fluorescently tagged proteins.

We estimated the localization precision using the following formula:

$$\sigma^2 = \left[\frac{s^2}{N} + \frac{\frac{a^2}{12}}{N} + \frac{8\pi s^4 b^4}{a^2 N^2}\right] \tag{2}$$

where $N$ is the number of photons collected which was on average 12.9 photons per 5-pixel window surrounding the centroid (data not shown); $s$ is the standard deviation of the microscope point-spread function, 294 nm; $a$ is the pixel size, 100 nm; and $b$ is the background intensity which was on average 0.8 photons per 5-pixel window. This results in a $\sigma = 82$ nm.

## Calculation of theoretical maximal hydrodynamic diffusion coefficients

The radius of gyration of SWR1 and Swc2 was calculated using the following formulas. First, the volume ($V$) of each particle was estimated using the following equation:

$$V\left(\text{nm}^3\right) = \frac{\left(\left(0.73\frac{\text{cm}^3}{\text{g}}\right)\left(10^{21}\frac{\text{nm}^3}{\text{cm}^3}\right)\right)}{6.023 * 10^{23}\frac{\text{Da}}{\text{g}}} * M\left(\text{Da}\right) \tag{3}$$

Then, the radius of gyration was estimated using the following equation:

$$R_{\min} = \left(\frac{3V}{4\pi}\right)^{\frac{1}{3}} \tag{4}$$

where $M$ is mass in Daltons (*Erickson, 2009*). Given the input of 1 MDa for SWR1 and 25.4 kDa for Swc2, the resulting radii of gyration are 6.62 nm SWR1 and 1.94 nm for Swc2. Next, the theoretical upper limit of 1D diffusion with no rotation was calculated using the following formula:

$$D = \frac{k_b T}{f} \tag{5}$$

where

$$f = 6\pi\eta R \tag{6}$$

**Table 3.** Summary of median diffusion coefficients as well as rejection criteria implemented per condition for particle refinement. Also included is information regarding biological and technical replicates. 'Trajs.' stands for trajectories.

| Condition DNA or Nuc array | Protein | Nucleotide | KCl (mM) | Biological replicates (BR) | Min. and Max # technical replicates (TR) per BR | Total Trajs. post refine | Criteria for linear fit cutoff | Total Trajs. pre refine | Median $D$ (μm²/s) | SEM*√(π/2) (μm²/s) |
|---|---|---|---|---|---|---|---|---|---|---|
| DNA | SWR1 | ATP | 70 | 4 | 4–6 | 462 | $p < 0.05$, $r^2 > 0.8$ | 555 | 0.024 | 0.001 |
| DNA | SWR1 | None | 70 | 4 | 4–7 | 245 | $p < 0.05$, $r^2 > 0.8$ | 345 | 0.013 | 0.002 |
| DNA | SWR1 | ATPγS | 70 | 3 | 4–13 | 283 | $p < 0.05$, $r^2 > 0.8$ | 367 | 0.026 | 0.002 |
| DNA | SWR1 | ADP | 70 | 3 | 5–12 | 313 | $p < 0.05$, $r^2 > 0.8$ | 476 | 0.011 | 0.002 |
| DNA | SWR1 | ATP | 25 | 1 | 9 | 157 | $p < 0.05$, $r^2 > 0.8$ | 171 | 0.015 | 0.001 |
| DNA | SWR1 | ATP | 200 | 1 | 8 | 131 | $p < 0.05$, $r^2 > 0.8$ | 136 | 0.041 | 0.003 |
| DNA | Swc2 | None | 25 | 1 | 9 | 152 | $p < 0.05$, $r^2 > 0.8$ | 200 | 0.719 | 0.069 |
| DNA | Swc2 | None | 70 | 1 | 8 | 115 | $p < 0.05$, $r^2 > 0.8$ | 143 | 1.038 | 0.088 |
| DNA | Swc2 | None | 150 | 1 | 10 | 79 | $p < 0.05$, $r^2 > 0.8$ | 98 | 1.549 | 0.125 |
| Nuc array | SWR1 | ATP | 70 | 4 | 4–5 | 101 | $p < 0.05$, $r^2 > 0.8$ | 301 | 0.009 | 0.003 |
| DNA | dCas9 | None | 70 | 3 | 6–12 | 44 | None | 44 | $2.7 \times 10^{-4}$ | $3.7 \times 10^{-4}$ |

and $\eta$ is the viscosity $9 \times 10^{-10}$ pN s/nm$^2$ (**Schurr, 1979**). The resulting upper limit without rotation for SWR1, is 36.7 µm$^2$/s and for Swc2 it is 125 µm$^2$/s. When computing the upper limit of 1D diffusion with rotation, the following formula considers the energy dissipation that comes from rotating while diffusing:

$$f = 6\pi\eta R + \left(\frac{2\pi}{10\text{BP}}\right)^2 \left[8\pi\eta R^3 + 6\pi\eta R \left(R_{\text{oc}}\right)^2\right] \tag{7}$$

where $R_{\text{oc}}$ is the distance between the center of mass of the DNA and the bound protein, and 10 BP is the length of one helical turn or 3.4 nm (**Ahmadi et al., 2018**; **Bagchi et al., 2008**; **Blainey et al., 2009**). Since we do not have structures of SWR1 or Swc2 bound to dsDNA alone, we report both the maximal and minimal value of the theoretical upper limit, where the minimal value corresponds to $R_{\text{oc}} = R$ and the maximal value corresponds to $R_{\text{oc}} = 0$. For SWR1 this minimum value is 0.105 µm$^2$/s and the maximum value is 0.183 µm$^2$/s whereas for Swc2 this minimum value is 4.01 µm$^2$/s and the maximum value is 6.86 µm$^2$/s.

## Scanning speed estimation

Lambda DNA tethered at its ends to two optically trapped beads was pulled to a tension of 5 pN, which resulted in a length approximately 92% of its contour length (15.2 µm). The length per basepair of DNA, 0.31 nm, is therefore slightly shorter than the value at full contour length (**Baumann et al., 2000**). The length of the NDR, 150 bp, in our conditions is therefore roughly 0.047 µm long. Since our localization precision is low, ~82 nm (see **Equation 2**), we do not have diffusion information at the resolution of basepairs, and therefore do not consider discrete models to approximate scanning speed. Given a median diffusion coefficient of SWR1 in the presence of 1 mM ATP of 0.024 µm$^2$/s, and the one-dimensional translational diffusion, $l = 2Dt$, where $l$ is the length in µm of DNA, we can approximate the time required to scan this length of DNA to be 0.093 s assuming a continuous model (**Berg, 1983**).

## Acknowledgements

We would like to thank Dr. Sina Jazani for helping validate our analysis, as well as by providing us with Hidden Markov analysis of diffusion of SWR1 in the presence and absence of nucleosomes. We thank Dr. Kelsey Bettridge for providing a template script for single-particle tracking in Matlab. This work was supported by the National Institutes of Health S10 OD025221 (cTrap grant, core-facilities JHU SOM), T.H. is an investigator of the Howard Hughes Medical Institute, an National Institutes of Health R35 GM122569 (to T.H.), an National Institutes of Health R01 GM125831 (to C.W.), a National Science Foundation, Graduate Research Fellowship Program DGE-1746891 (to C.C.C.), an National Institutes of Health training grant T32 GM007445 (to C.C.C.), an National Institutes of Health Postdoctoral Training Fellowship F32 GM128299 (to M.F.P.), and an NIGMS, F32GM133151 (to R.K.L).

## Additional information

### Funding

| Funder | Grant reference number | Author |
| --- | --- | --- |
| National Institutes of Health | S10 OD025221 | Taekjip Ha |
| Howard Hughes Medical Institute | | Taekjip Ha |
| National Institutes of Health | R35 GM122569 | Taekjip Ha |
| National Institutes of Health | R01 GM125831 | Carl Wu |
| National Science Foundation | DGE-1746891 | Claudia C Carcamo |

| Funder | Grant reference number | Author |
|---|---|---|
| National Institutes of Health | T32 GM007445 | Claudia C Carcamo |
| National Institutes of Health | F32 GM128299 | Matthew F Poyton |
| National Institutes of Health | F32 GM133151 | Robert K Louder |

The funders had no role in study design, data collection, and interpretation, or the decision to submit the work for publication.

## Author contributions

Claudia C Carcamo, Conceptualization, Resources, Data curation, Software, Formal analysis, Funding acquisition, Validation, Investigation, Visualization, Methodology, Writing - original draft, Project administration; Matthew F Poyton, Conceptualization, Resources, Data curation, Software, Formal analysis, Funding acquisition, Validation, Investigation, Visualization, Methodology, Writing - original draft, Writing – review and editing; Anand Ranjan, Resources, Writing – review and editing; Giho Park, Resources; Robert K Louder, Resources, Funding acquisition; Xinyu A Feng, Formal analysis, Investigation; Jee Min Kim, Formal analysis; Thuc Dzu, Investigation; Carl Wu, Taekjip Ha, Supervision, Funding acquisition, Project administration, Writing – review and editing

## Author ORCIDs

Claudia C Carcamo ◉ http://orcid.org/0000-0002-2646-188X
Matthew F Poyton ◉ http://orcid.org/0000-0003-1261-2138
Anand Ranjan ◉ http://orcid.org/0000-0001-6071-6017
Carl Wu ◉ http://orcid.org/0000-0001-6933-5763
Taekjip Ha ◉ http://orcid.org/0000-0003-2195-6258

## Decision letter and Author response

Decision letter https://doi.org/10.7554/eLife.77352.sa1
Author response https://doi.org/10.7554/eLife.77352.sa2

# Additional files

## Supplementary files

• Transparent reporting form

## Data availability

Raw data have been uploaded to Dryad in the form of a Matlab structured arrays. All Matlab codes used to generate the main figures are publicly available at https://github.com/ccarcam1/SWR1_1D_Diffusion_Publication (copy archived at swh:1:rev:2da897b5428e121eccf7a05bfb93c88928aafb02).

The following dataset was generated:

| Author(s) | Year | Dataset title | Dataset URL | Database and Identifier |
|---|---|---|---|---|
| Carcamo CC | 2022 | Data from: ATP Binding Facilitates Target Search of SWR1 Chromatin Remodeler by Promoting One-Dimensional Diffusion on DNA | https://doi.org/10.5061/dryad.ghx3ffbqw | Dryad Digital Repository, 10.5061/dryad.ghx3ffbqw |

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
