## [Editor Report]

This manuscript provides exciting and timely new insight into the target search mechanism of SWR1, with fundamentally important implications for other nucleosome remodelers. The authors use extremely elegant single-molecule approaches to study the interaction of SWR1 with DNA. This study provides the first conclusive demonstration that SWR1 undergoes 1D diffusion along DNA, which plays an important role in finding the correct nucleosomal substrate in vivo by guiding SWR1 molecules that bind to nucleosome-depleted regions towards flanking nucleosomes.

---

## [Decision Letter]

**Decision letter after peer review:**

Thank you for submitting your article "ATP Binding Facilitates Target Search of SWR1 Chromatin Remodeler by Promoting One-Dimensional Diffusion on DNA" for consideration by *eLife*. Your article has been reviewed by 3 peer reviewers, including Jerry L Workman as the Reviewing Editor and Reviewer #1, and the evaluation has been overseen by Kevin Struhl as the Senior Editor. The following individuals involved in review of your submission have agreed to reveal their identity: Sebastian Deindl (Reviewer #2); Tom Owen-Hughes (Reviewer #3).

Essential revisions:

Please note that only textual changes are required, but you may consider including additional data as indicated below. In addition, clarifications in the text that point out the limitations of the study and that not all of SWR1 function (i.e. directionality of H2AZ incorporation) is explained are indicated below.

1) The effect of ATP on the diffusion is very intriguing and strongly suggests the involvement of the Swr1 subunit. The authors could consider analyzing the diffusion of the motor subunit Swr1. This should enable a distinction between the direct change in sliding properties of the Swr1 helicase domains vs the allosteric effect on the Swc2 conformation.

2) In the presence of nucleosomes, specific binding of SWR1 should occur. More specifically, one could expect SWR1 to first bind randomly and start diffusing before finally encountering a nucleosome and forming a long-lived complex. Were the authors able to observe it? It would be very helpful if the authors added a more detailed description of the SWR1 behavior they observed in the presence of nucleosomes, including a quantification of immobile SWR1 molecules as a function of time after initial binding to DNA – the expectation would be that the fraction of immobile SWR1 molecules would increase due to nucleosome binding. The authors could strengthen the manuscript if they conducted SWR1 diffusion experiments in the presence of labeled nucleosomes, which should help answer these questions and enable more confident conclusions.

3) Lines 349-351: "Λ DNA, the DNA substrate used in this study, has asymmetric AT-content, which has been shown to affect nucleosome positioning during random deposition" – could the authors plot the distribution of SWR1 diffusion coefficients as a function of the position on the substrate? Even when the λ-DNA orientation is not known, the effect of sequence (e.g. AT-content) on diffusion should be apparent in such a plot.

4) The authors may consider discussing in more quantitative terms the magnitude of the effect that the 1D search process could have on directing SWR1 towards +1 nucleosomes. It would seem that a comparison with -1 nucleosomes, where 1D diffusion is expected to have the same effect as for +1, could be rather informative.

5) The alternative explanation for an increased dissociation of SWR1 in the presence of competitor DNA could be a "monkey bar" mechanism, where multiple DNA binding sites can engage in interactions with different DNA molecules, thus facilitating dissociation. The authors could probe this hypothesis by measuring toff for Swc2 with and without competitor DNA. If the "monkey-bar" mechanism were true, Swc2 dissociation should not be significantly affected by competitor DNA, since it's only one of several DNA binding domains of SWR1, whereas if the original effect was caused by hopping, competitor DNA should still significantly enhance Swc2 dissociation.

6) It's clear that the association of SWR1 complexes with DNA is affected by the length of the fragment. The major limitation of the study is that it does not establish whether these properties are required for correct targeting activity in vivo.

7) It's much more difficult to test this, and likely beyond the scope of this study. One way might be to identify mutations that affect the diffusive motion of SWR1 complexes but not their activity in histone exchange. In the absence of data of this nature it is important the manuscript acknowledges that further work would be required to establish that the biophysical properties reported here in vitro are rate limiting for SWR1 action in vivo.

8) Other issues that appear not to be addressed by the model include that there is not a universally accepted view on what factors are bound to f NDR's. Some studies have suggested that fragile nuclease sensitive sub nucleosomes are present within NDR's other that transcription factors are dynamically associated with these regions. The presence of either would affect the effective length in the NDR and reduce its ability to influence SWR1 targeting via the mechanism proposed here.

9) DNA fragment length is observed to have a larger effect on the off rate of SWR1 complexes (300-fold), than the on rate (30-fold). If SWR1 complexes are stably associated with longer regions of free DNA, this may impede their association with nucleosomes where the complex acts to incorporate H2AZ.

10) Although the presence of relatively accessible DNA may explain some aspects of the targeting of SWR1 complexes, it does not explain why H2AZ is incorporated into +1 nucleosomes to a greater extent than -1 nucleosomes. As a result the one dimensional diffusion model is unable to explain all aspects of SWR1 targeting.

11) Limitations of the biophysical measurements reported in the manuscript in the absence of corroborative evidence relating to SWR1 function in cells should be included in the discussion.

12. While it would require additional methods and perhaps beyond the scope of the current studies addressing the following questions could increase the impact of the paper. Can the authors show that diffusion by SWR1 actually enhances H2AZ exchange into nucleosomes? The directionality of H2AZ insertion (+1 vs -1 nucleosomes) is an outstanding question that the authors only elude to in the discussion. Can the authors show that histone acetylation anchors SWR1 to the +1 nucleosome or show that acetylated nucleosomes are preferentially exchanged with H2AZ? Such experiments would support their suggestion regarding directionality.

---

## [Author Response]

Essential revisions:Please note that only textual changes are required, but you may consider including additional data as indicated below. In addition, clarifications in the text that point out the limitations of the study and that not all of SWR1 function (i.e. directionality of H2AZ incorporation) is explained are indicated below.

We have performed new experiments, extended our data analysis, and made substantial modifications to the manuscript to carefully address all the points raised. The revised manuscript includes a more thorough discussion on the limitations of our in vitro diffusion measurement on fully explaining in vivo SWR1 behavior and function. Moreover, we performed additional TIRF measurements to address how Swc2 binding to dsDNA is affected by competitor DNA. Finally, we made the corrections noted in the minor points section and we have performed the suggested supplemental data analyses. The resulting manuscript is a stronger and more complete work, which we believe is suitable for publication in *eLife*.

1) The effect of ATP on the diffusion is very intriguing and strongly suggests the involvement of the Swr1 subunit. The authors could consider analyzing the diffusion of the motor subunit Swr1. This should enable a distinction between the direct change in sliding properties of the Swr1 helicase domains vs the allosteric effect on the Swc2 conformation.

We agree that directly observing the ATP-binding motor subunit, Swr1, diffusion on DNA would help disentangle contributions made to the overall diffusive behavior of the SWR1 complex by the Swr1 subunit vs the Swc2 subunit.

However, the Swr1 subunit is structurally intertwined with the Rvb ring, which suggests that it would be unstable in isolation (Willhoft et al., 2018). Indeed, the Wu lab has noted that the Swr1 subunit aggregates when expressed alone. Thus, more complex schemes are necessary to investigate whether ATP-binding increases diffusion of the complex by direct conformational changes of Swr1 and/or by allosteric effects on the conformation of Swc2. This is an interesting problem for future investigations and is now noted in the first paragraph of the second section of the Discussion.

“Future studies will aim to address how ATP-binding to the catalytic subunit may alter its conformation and/or that of Swc2 (or another DNA binding component) leading to enhanced diffusion of the SWR1 complex.”

2) In the presence of nucleosomes, specific binding of SWR1 should occur. More specifically, one could expect SWR1 to first bind randomly and start diffusing before finally encountering a nucleosome and forming a long-lived complex. Were the authors able to observe it? It would be very helpful if the authors added a more detailed description of the SWR1 behavior they observed in the presence of nucleosomes, including a quantification of immobile SWR1 molecules as a function of time after initial binding to DNA – the expectation would be that the fraction of immobile SWR1 molecules would increase due to nucleosome binding. The authors could strengthen the manuscript if they conducted SWR1 diffusion experiments in the presence of labeled nucleosomes, which should help answer these questions and enable more confident conclusions.

We thank the reviewers for this suggestion. We have performed instantaneous diffusion analysis of SWR1 diffusion on naked DNA, on nucleosome arrays, as well as for dCas9, and used this to explore the predictions mentioned by the reviewers. Due to technical considerations for all of our optical tweezer experiments, we are unable to resolve the initial nucleosome binding event as most SWR1 particles were already bound to the array and exhibiting a nucleosome-captured state at the start of imaging. This is best seen in Figure 6 —figure supplement 2B, which shows that the mean instantaneous diffusion coefficient doesn’t decrease over time after the start of the trace observation, nor does the percentage of all traces in an immobile state increase over time. We agree that using labeled nucleosomes would strengthen our claims on SWR1 diffusion in the presence of nucleosomes; however, we are still in the process of optimizing a long-lived fluorescent label on the nucleosome for the optical tweezers set-up.

The results of this analysis are now included in Figure 6 —figure supplement 2. Our interpretations are now found as the third paragraph in the Results section entitled “Nucleosomes are barriers to SWR1 diffusion”.

In addition to the new analysis, we also added two panels to Figure 6 —figure supplement 1 (D-E) which demonstrate that the diffusive behavior of SWR1 on the nucleosome array is confined, further strengthening the claims initially presented.

3) Lines 349-351: "Λ DNA, the DNA substrate used in this study, has asymmetric AT-content, which has been shown to affect nucleosome positioning during random deposition" – could the authors plot the distribution of SWR1 diffusion coefficients as a function of the position on the substrate? Even when the λ-DNA orientation is not known, the effect of sequence (e.g. AT-content) on diffusion should be apparent in such a plot.

We thank the reviewers for raising this important point. Since we did not collect the data in a way that would ensure that we could correlate the position of the particle with the underlying sequence of the DNA, we cannot provide conclusive evidence for a correlation between diffusion coefficient and underlying AT-content. In order to address the reviewer’s request, we plotted the mean instantaneous diffusion coefficient as a function of the position along the substrate (measured in pixels in Author response image 1) and found that the diffusion coefficient does not significantly vary as a function of position (the differences between positions fall within the standard deviation of the mean of all other sites on the substrate). Note that the AT content, averaged over two different DNA orientations, does show a slight peak in the center of the DNA. Based on this analysis, the diffusion coefficients do not display a correlated or anticorrelated relationship with AT-content.

We believe that future work should address the possibility that there is a difference in mobility of the SWR1 remodeler on DNA based on AT-content. This could be accomplished by using fiducial markers on the DNA to better align the DNA sequence with the position of the bound diffusing chromatin remodeler. Another possibility is to create a sequence with well-defined repeats of biologically relevant sites of high vs low AT content. Yeast promoters are known to be higher in AT-content and a model promoter/ NDR might serve as a better substrate for studying differences in diffusion based on underlying sequence composition. It is possible that the rigidity of such sequences may modulate the free energy barrier required for 1D scanning of these regions of open chromatin resulting in difference in mobility of the remodeler.

**Author response image 1. sa2fig1:** Changes in instantaneous diffusion coefficient as a function of position along the DNA substrate. (A) The AT-content along the Λ phage genome is plotted in both orientations, revealing an asymmetric distribution of AT-content directionally across the genome. Averaging the AT-content across the two orientations shows that the center of the DNA has a slightly higher AT-content, indicating that if there is any sequence correlation to SWR1 diffusion, we may see a difference in mean diffusion coefficients at the center of the DNA (since the DNAs in this study were not intentionally oriented). (B) The instantaneous diffusion coefficient of SWR1 as a function of position along the DNA substrate. The center of the DNA substrate does not appear to exhibit any enrichment for higher or lower mobility in the SWR1 remodeler. Nonetheless, the experimental set-up would need to be further optimized to determine if there is in fact no correlation with sequence or if there is simply not enough resolution in our current set-up to discern one.

4) The authors may consider discussing in more quantitative terms the magnitude of the effect that the 1D search process could have on directing SWR1 towards +1 nucleosomes. It would seem that a comparison with -1 nucleosomes, where 1D diffusion is expected to have the same effect as for +1, could be rather informative.

This is an important point that deserved more attention in the manuscript. We have added the following paragraphs to the first part of the Discussion section “Reducing the dimensionality of nucleosome target search”, which we believe addresses the concerns raised by the reviewer:

“While this study establishes for the first time that the chromatin remodeler SWR1 can diffuse on DNA, and that such diffusion is confined by dCas9 and nucleosomes, many questions remain. First, while SWR1 may be guided to its main target nucleosome, +1, via binding to the NDR, how SWR1 preferentially recognizes the +1 nucleosome over the -1 nucleosome on the other side of the NDR remains unanswered. While there is evidence that acetylation of the +1 nucleosome may help establish the directionality of the interaction when assessing the two NDR flanking nucleosomes (Ranjan et al., 2013; Zhang et al., 2005), data from our study indicates that even nucleosomes lacking acetylation can reduce the diffusion of the SWR1 complex possibly through nucleosome engagement. Therefore, it will be necessary to measure the fold difference in dwell time on nucleosomes, in the context of 1D diffusion, that do and do not contain acetylation marks in order to determine the magnitude of the effect that the 1D search process would have on directing binding to the +1 vs -1 nucleosome. Of note, there are times where both NDR flanking nucleosomes will be substrates of SWR1 and are equally enriched for H2A.Z incorporation, as is the case when both nucleosomes are the +1 nucleosomes of divergently transcribed genes (Bagchi et al., 2020). Additionally, H2A.Z is preferentially enriched at inactive promoters (Li et al., 2005); this phenomenon should be revisited in the light of 1D diffusion as it is possible that regulatory mechanisms that affect transcription of these genes also impact the ease with which SWR1 can perform 1D diffusion.

Furthermore, the DNA used as a substrate in this study, λ DNA, is different in mechanical properties, owing to differences in sequence composition, compared to yeast promoters and may therefore not be the most biologically representative substrate for studying 1D diffusion of SWR1. A sequencing study discovering the intrinsic flexibility of different DNA sequences helped establish that promoter NDR regions from yeast are highly rigid and this unique property compared to gene body DNA helps instruct the chromatin remodeler, INO80, a close relative of SWR1, to position nucleosomes at the boundary of the rigid NDR helping to establish the defined positioning of the +1 nucleosome (Basu et al., 2021). It is possible that sensing changes in DNA rigidity while performing a 1D search could allow SWR1 to distinguish +1 from -1 nucleosomes. Finally, in our study, a distinction is not made between nucleosome depleted regions (NDRs) and nucleosome free regions (NFRs) in contemplating the relevance of a 1D search mechanism in targeting SWR1 to the +1 nucleosome. The distinction between these two types of promoter architectures was recently further developed via the implementation of ChIP–exo/seq to comprehensively discover occupancies of fragile nucleosome in addition to bound transcription factors at yeast promoters (Rossi et al., 2021). The complexity of the integrated transcriptional regulatory networks and 3D chromatin architecture is a caveat to the simplified model presented in this study and may limit the magnitude to which a 1D search process factors into target nucleosome binding in vivo. Fragile nucleosomes in NDRs effectively shorten the length of nucleosome free linker DNA proximal to the +1 nucleosome, diminishing the kinetic binding preference for that NDR. Fragile nucleosomes, in addition to transcription factors, may also impede diffusion, increasing the timescale required to encounter flanking nucleosomes via 1D diffusion. in vivo binding of SWR1, as shown via ChIP-seq, has already been shown to be reduced by fragile nucleosome occupancy of NDRs (Ranjan *et al.*, 2013; Venters and Pugh, 2009). Therefore, while the observations presented in this study are an exciting launching point into studies of chromatin remodeler 1D diffusion, future work with additional promoter-binding proteins is needed to assess the relevance of these biophysical measurements to in vivo target binding.”

5) The alternative explanation for an increased dissociation of SWR1 in the presence of competitor DNA could be a "monkey bar" mechanism, where multiple DNA binding sites can engage in interactions with different DNA molecules, thus facilitating dissociation. The authors could probe this hypothesis by measuring toff for Swc2 with and without competitor DNA. If the "monkey-bar" mechanism were true, Swc2 dissociation should not be significantly affected by competitor DNA, since it's only one of several DNA binding domains of SWR1, whereas if the original effect was caused by hopping, competitor DNA should still significantly enhance Swc2 dissociation.

This was a very interesting suggestion. We decided to probe this hypothesis by measuring t_bound_ for Swc2 with and without competitor DNA (as well as in the presence of high salt) as the reviewer suggested and observed that Swc2 DBD alone experiences shortened binding lifetime in the presence of DNA. We did not rule out the possibility of SWR1 and Swc2 utilizing a monkey-bars mechanism since the DNA binding domain of Swc2 is likely disordered and may be able to make interactions with multiple pieces of DNA simultaneously inducing a monkey-bar exchange. In addition to appending the additional data to Figure 1 —figure supplement 1G-H, we added the following to the Results in the section discussing SWR1 and Swc2 hopping versus sliding:

“We also observed the effects of competitor DNA on Swc2 DNA binding domain dwell-time under the same conditions (Figure 1 —figure supplement 1G-H). Like SWR1, Swc2 DBD experienced shortened dwell-times in the presence of competitor DNA (Figure 1 —figure supplement 1H). This combined with the observation that high salt decreases the binding lifetime of Swc2 in the context of TIRF imaging further supports that Swc2 and SWR1 engage in hopping during diffusion. An alternative explanation for why competitor DNA shortens binding lifetime is that both SWR1 and Swc2 DBD may engage in a so-called “monkey-bars” mechanism in which binding at a second DNA binding site promotes dissociation from DNA at the first bound site (Rudolph et al., 2018). This mechanism cannot be ruled out by our observations since Swc2 DBD is likely intrinsically disordered and may be able to bind multiple DNA simultaneously similar to how distant DNA binding domains on the SWR1 complex may promote exchange onto competitor DNA.”

6) It's clear that the association of SWR1 complexes with DNA is affected by the length of the fragment. The major limitation of the study is that it does not establish whether these properties are required for correct targeting activity in vivo.

Although the reviewer is correct, there is prior literature reporting that SWR1, in vivo*,* accumulates at promoters with longer nucleosome depleted regions (NDRs), as seen by SWR1 chromatin immunoprecipitation (ChIP), (Ranjan *et al.*, 2013; Venters and Pugh, 2009). The referenced study used partial and full MNase digestions showing that the fragile nucleosome occupancy of some NDRs affects the SWR1 occupancy trend; fragile nucleosomes in the middle of long NDRs effectively shortens the region of long nucleosome free DNA resulting in decreased SWR1 targeting to these promoters. In our initial submission, we left this unaddressed in the text, and have further clarified these points in the third paragraph of the first section of the discussion titled “Reducing the dimensionality of nucleosome target search”.

7) It's much more difficult to test this, and likely beyond the scope of this study. One way might be to identify mutations that affect the diffusive motion of SWR1 complexes but not their activity in histone exchange. In the absence of data of this nature it is important the manuscript acknowledges that further work would be required to establish that the biophysical properties reported here in vitro are rate limiting for SWR1 action in vivo.

We thank the reviewers for pointing this out and have added the potential usefulness of discovering such mutants to our conclusions section:

“They [*in reference to future studies*] might also include an investigation of mutations to SWR1 subunits that permit histone exchange but abrogate diffusivity on DNA, which would establish the importance of this biophysical property to SWR1 activity in vivo.”

8) Other issues that appear not to be addressed by the model include that there is not a universally accepted view on what factors are bound to f NDR's. Some studies have suggested that fragile nuclease sensitive sub nucleosomes are present within NDR's other that transcription factors are dynamically associated with these regions. The presence of either would affect the effective length in the NDR and reduce its ability to influence SWR1 targeting via the mechanism proposed here.

The reviewer is correct. We have cited the referenced paper that better characterized the occupancy of fragile nucleosomes and TFs at yeast NDRs, and have mentioned the caveats that such in vivo complexity of integrated transcription regulatory networks would pose to the simplified model we present in this study based on in vitro observations. The following was added to the third paragraph of the Discussion section titled “Reducing the dimensionality of nucleosome target search”:

“Finally, in our study, a distinction is not made between nucleosome depleted regions (NDRs) and nucleosome free regions (NFRs) in contemplating the relevance of a 1D search mechanism in targeting SWR1 to the +1 nucleosome. The distinction between these two types of promoter architectures was recently further developed via the implementation of ChIP–exo/seq to comprehensively discover occupancies of fragile nucleosome in addition to bound transcription factors at yeast promoters (Rossi *et al.*, 2021). The complexity of the integrated transcriptional regulatory networks and 3D chromatin architecture is a caveat to the simplified model presented in this study and may limit the magnitude to which a 1D search process factors into target nucleosome binding in vivo. Fragile nucleosomes in NDRs effectively shorten the length of nucleosome free linker DNA proximal to the +1 nucleosome, diminishing the kinetic binding preference for that NDR. Fragile nucleosomes, in addition to transcription factors, may also impede diffusion, increasing the timescale required to encounter flanking nucleosomes via 1D diffusion. in vivo binding of SWR1, as shown via ChIP-seq, has already been shown to be reduced by fragile nucleosome occupancy of NDRs (Ranjan *et al.*, 2013; Venters and Pugh, 2009). Therefore, while the observations presented in this study are an exciting launching point into studies of chromatin remodeler 1D diffusion, future work with additional promoter-binding proteins is needed to assess the relevance of these biophysical measurements to in vivo target binding.”

9) DNA fragment length is observed to have a larger effect on the off rate of SWR1 complexes (300-fold), than the on rate (30-fold). If SWR1 complexes are stably associated with longer regions of free DNA, this may impede their association with nucleosomes where the complex acts to incorporate H2AZ.

We thank the reviewers for pointing this out. We have changed the text to clarify our understanding that the increase in on-rate as a function of DNA length may in fact be best translated to understanding SWR1 targeting to +1 nucleosomes in vivo, and that the decrease in off-rate with DNA length, seen in vitro, is more likely to be affected by the cellular environment in vivo. Ultimately, the photobleaching-limited lifetime of SWR1 binding to DNA in our set-up allowed us to make an extended observation time of single diffusing SWR1 particles, enabling the characterization of the biophysical properties of SWR1 1D diffusion using the cTrap. We have added the following to the end of the first Results section alongside the presentation of the TIRF data:

“Measurements at lower laser power showed that SWR1 remained bound to 150 bp DNA with a half-life of approximately 3 minutes (Figure 1—figure supplement 1C). *t*_bound_ was unchanged in the presence of ATP but was sensitive to ionic strength, decreasing with added salt (Figure 1—figure supplement 1E-F). Curiously, *t*_bound_ also decreased in the presence of competitor DNA (Figure 1—figure supplement 1E-F). In our in vitro experimental set-up, as DNA length is increased, SWR1 t-bound showed an approximately 120-fold increase. Compared to the 36-fold increase in k-bind as DNA length increased, the much higher fold increase in binding lifetime might suggest that once SWR1 binds NDR DNA in vivo that it will remain bound for several minutes, potentially sequestering the remodeler from performing histone exchange at other +1 nucleosome targets. The in vivo t-bound, however, is likely much shorter due to the higher ionic strength in cells as well as due to effects of molecular crowding, competitor DNA binding, and the activity of endogenous helicases which may oust DNA bound factors such as SWR1. Indeed, a study utilizing single particle tracking in vivo showed that the stable chromatin bound dwell-time for a number of ATP-dependent remodelers is on the order of several seconds (Kim et al., 2021).”

10) Although the presence of relatively accessible DNA may explain some aspects of the targeting of SWR1 complexes, it does not explain why H2AZ is incorporated into +1 nucleosomes to a greater extent than -1 nucleosomes. As a result the one dimensional diffusion model is unable to explain all aspects of SWR1 targeting.

The reviewer is correct. We have clarified this point in the second and third paragraph of the first section of the discussion titled “Reducing the dimensionality of nucleosome target search” as well as in the concluding paragraph.

11) Limitations of the biophysical measurements reported in the manuscript in the absence of corroborative evidence relating to SWR1 function in cells should be included in the discussion.

We agree with the reviewers. However, previous studies have shown that SWR1 diffusion when bound to chromatin is larger than the diffusion of H2B incorporated on chromatin (Ranjan et al., 2020); this was also comprehensively shown for many other yeast chromatin remodelers in a separate study (Kim *et al.*, 2021). We have added this information to the Discussion section and have also stressed at the reviewers’ suggestion that without using mutants of SWR1 which are defective for DNA binding/ DNA scanning, we do not have in vivo cell data that demonstrates the essentiality of DNA binding/ DNA scanning to SWR1 function. Such data would indicate the magnitude to which SWR1 DNA binding and scanning affects H2A.Z incorporation in vivo.

The following was added to the concluding paragraph:

“While our study lacks cellular data demonstrating the essentiality of 1D scanning for SWR1 target localization efficiency and SWR1 function in vivo, SWR1 bound to chromatin has been shown to have a bound state diffusion coefficient that is ~2-fold larger than the diffusion of chromatin itself (SWR1 D bound 0.063 +/- 0.0003 μm^2^/sec versus histone H2B D bound 0.028+/- 0.0002 μm^2^/sec) (Ranjan *et al.*, 2020). A later study found that other yeast chromatin remodelers that act on nucleosomes adjacent to the NDR (RSC, SWI/SNF, ISW2, and INO80) are similarly more diffusive than histone H2B in their chromatin-bound states, and are, similar to SWR1, dependent on the ability of their ATPase domains to bind or hydrolyze ATP in order to exhibit this enhanced bound diffusion (Kim *et al.*, 2021). While several factors could result in this enhanced bound diffusion, such as enhanced diffusion of promoter chromatin or intersegmental transfer of SWR1 (a topic not explored in our study), 1D diffusion may be an important contributor to this enhanced chromatin bound diffusion.”

12. While it would require additional methods and perhaps beyond the scope of the current studies addressing the following questions could increase the impact of the paper. Can the authors show that diffusion by SWR1 actually enhances H2AZ exchange into nucleosomes? The directionality of H2AZ insertion (+1 vs -1 nucleosomes) is an outstanding question that the authors only elude to in the discussion. Can the authors show that histone acetylation anchors SWR1 to the +1 nucleosome or show that acetylated nucleosomes are preferentially exchanged with H2AZ? Such experiments would support their suggestion regarding directionality.

We thank the reviewers for their comments which foreshadow future directions for this project involving the use of acetylated nucleosomes reconstituted on λ DNA on the optical trap combined with 3-color nucleosome FRET to detect histone exchange on encounter by SWR1 (Poyton et al. 2022). Note that it has already been shown that, in a competition assay using dinucleosomes with linker DNA the length of a typical yeast NDR, SWR1 preferentially exchanges acetylated nucleosomes over unacetylated nucleosomes [See Figure 5, panel B from the referenced study] (Ranjan *et al.*, 2013).

Our labs are in the process of optimizing fluorescently labeled nucleosomes which can be used to determine if acetylated nucleosomes anchor SWR1 binding, as suggested by the reviewers. We are also in the process of optimizing an instantaneous diffusion analysis pipeline required to perform this dwell-time analysis.

We have revised the conclusion to clarify these points by adding:

“While 1D diffusion should in principle allow SWR1 to encounter either the +1 or -1 nucleosome at the ends of an NDR, we anticipate that future studies of diffusion towards acetylated nucleosomes may provide insights on how the complex is preferentially enriched at the +1 nucleosome that bears a higher level of histone acetylation.

References:

Bagchi, D.N., Battenhouse, A.M., Park, D., and Iyer, V.R. (2020). The histone variant H2A.Z in yeast is almost exclusively incorporated into the +1 nucleosome in the direction of transcription. Nucleic Acids Res *48*, 157-170. 10.1093/nar/gkz1075.

Basu, A., Bobrovnikov, D.G., Qureshi, Z., Kayikcioglu, T., Ngo, T.T.M., Ranjan, A., Eustermann, S., Cieza, B., Morgan, M.T., Hejna, M., et al. (2021). Measuring DNA mechanics on the genome scale. Nature *589*, 462-467. 10.1038/s41586-020-03052-3.

Kim, J.M., Visanpattanasin, P., Jou, V., Liu, S., Tang, X., Zheng, Q., Li, K.Y., Snedeker, J., Lavis, L.D., Lionnet, T., and Wu, C. (2021). Single-molecule imaging of chromatin remodelers reveals role of ATPase in promoting fast kinetics of target search and dissociation from chromatin. *eLife 10*. 10.7554/*eLife*.69387.

Li, B., Pattenden, S.G., Lee, D., Gutierrez, J., Chen, J., Seidel, C., Gerton, J., and Workman, J.L. (2005). Preferential occupancy of histone variant H2AZ at inactive promoters influences local histone modifications and chromatin remodeling. Proc Natl Acad Sci U S A *102*, 18385-18390. 10.1073/pnas.0507975102.

Ranjan, A., Mizuguchi, G., FitzGerald, P.C., Wei, D., Wang, F., Huang, Y., Luk, E., Woodcock, C.L., and Wu, C. (2013). Nucleosome-free region dominates histone acetylation in targeting SWR1 to promoters for H2A.Z replacement. Cell *154*, 1232-1245. 10.1016/j.cell.2013.08.005.

Ranjan, A., Nguyen, V.Q., Liu, S., Wisniewski, J., Kim, J.M., Tang, X., Mizuguchi, G., Elalaoui, E., Nickels, T.J., Jou, V., et al. (2020). Live-cell single particle imaging reveals the role of RNA polymerase II in histone H2A.Z eviction. *ELife 9*. 10.7554/*eLife*.55667.

Rossi, M.J., Kuntala, P.K., Lai, W.K.M., Yamada, N., Badjatia, N., Mittal, C., Kuzu, G., Bocklund, K., Farrell, N.P., Blanda, T.R., et al. (2021). A high-resolution protein architecture of the budding yeast genome. Nature *592*, 309-314. 10.1038/s41586-021-03314-8.

Rudolph, J., Mahadevan, J., Dyer, P., and Luger, K. (2018). Poly(ADP-ribose) polymerase 1 searches DNA via a ‘monkey bar’ mechanism. *eLife 7*. 10.7554/*eLife*.37818.

Venters, B.J., and Pugh, B.F. (2009). A canonical promoter organization of the transcription machinery and its regulators in the Saccharomyces genome. Genome Res *19*, 360-371. 10.1101/gr.084970.108.

Willhoft, O., Ghoneim, M., Lin, C.L., Chua, E.Y.D., Wilkinson, M., Chaban, Y., Ayala, R., McCormack, E.A., Ocloo, L., Rueda, D.S., and Wigley, D.B. (2018). Structure and dynamics of the yeast SWR1-nucleosome complex. Science *362*. 10.1126/science.aat7716.

Zhang, H., Roberts, D.N., and Cairns, B.R. (2005). Genome-wide dynamics of Htz1, a histone H2A variant that poises repressed/basal promoters for activation through histone loss. Cell *123*, 219-231. 10.1016/j.cell.2005.08.036.